# 4E Assessment of an Organic Rankine Cycle (ORC) Activated with Waste Heat of a Flash–Binary Geothermal Power Plant

**DOI:** 10.3390/e24121832

**Published:** 2022-12-15

**Authors:** Víctor M. Ambriz-Díaz, Israel Y. Rosas, Oscar Chávez, Carlos Rubio-Maya

**Affiliations:** 1Tecnológico Nacional de México/I. T. Chihuahua, Av. Tecnológico, 2909, Chihuahua 31310, Chihuahua, Mexico; 2Faculty of Mechanical Engineering, “W” Building, Central Campus, Universidad Michoacana de San Nicolas de Hidalgo, Morelia 58030, Michoacán, Mexico

**Keywords:** flash–binary power plant, organic Rankine cycle, waste heat, exergoeconomic, exergoenvironmental, 4E assessment

## Abstract

In this paper, the 4E assessment (Energetic, Exergetic, Exergoeconomic and Exergoenvironmental) of a low-temperature ORC activated by two different alternatives is presented. The first alternative (S1) contemplates the activation of the ORC through the recovery of waste heat from a flash–binary geothermal power plant. The second alternative (S2) contemplates the activation of the ORC using direct heat from a geothermal well. For both alternatives, the energetic and exergetic models were established. At the same time, the economic and environmental impact models were developed. Finally, based on the combination of the exergy concepts and the economic and ecological indicators, the exergoeconomic and exergoenvironmental performances of the ORC were obtained. The results show higher economic, exergoeconomic and exergoenvironmental profitability for S1. Besides, for the alternative S1, the ORC cycle has an acceptable economic profitability for a net power of 358.4 kW at a temperature of 110 °C, while for S2, this profitability starts being attractive for a power 2.65 times greater than S1 and with a temperature higher than 135 °C. In conclusion, the above represents an area of opportunity and a considerable advantage for the implementation of the ORC in the recovery of waste heat from flash–binary geothermal power plants.

## 1. Introduction

Geothermal energy is available in temperatures ranging from 50 to 350 °C [1]. Within this range, various categories and uses of geothermal energy can be distinguished depending on the temperature. For example, high temperature geothermal energy (T > 150 °C) is mainly used for the production of electricity through single or double flash geothermal power plants, medium temperature geothermal energy (100–150 °C) is mainly used for the production of electricity through the Organic Rankine Cycle and the Kalina cycle (KAC), and low-temperature geothermal energy (T < 100 °C) is used in the so-called direct uses and recently in the generation of electricity through the use of low-temperature ORC technology [2]. Although the use of low-temperature geothermal energy is not considered feasible in the generation of electricity on an individual basis, ORC technology is very attractive when it comes to the recovery of low-temperature waste heat in geothermal plants [3]. In this way, flash–binary geothermal power plants are distinguished, in which the waste heat of the flash geothermal power plant is used as an activation resource for a binary plant to substantially increase energy production [4]. In the same way, it contributes to harnessing the geothermal resource in the best possible way and lower reinjection temperatures are also reached. Within the binary plants coupled to geothermal power plants, the ORC stands out for achieving good thermodynamic and economic performance compared to the existing binary cycle technologies (Kalina cycle and Goswami cycle) [5]. It is due to the above that the use of waste heat in geothermal power plants by implementing low-temperature ORC technology is one of the most promising strategies. Furthermore, the proposal of new integrated cycles, based on well-known thermodynamic cycles, is an effective way to improve the performance of traditional cycles. In recent decades, the integration of different cycles for simultaneous power production has attracted great attention, for example, by using multiple flash–binary power plants. These types of systems allow the efficient use of primary energy resources, as well as increase the thermal efficiency of the systems and reduce the emission of greenhouse gases [6].

Responding to the exposed situation, the scientific community has focused on investigating this type of geothermal power plants from thermodynamic perspectives (exergetic, exergoeconomic and exergoenvironmental). From a thermodynamic point of view, exergetic analysis has proven to be a powerful tool for identifying the location, magnitude, and sources of thermodynamic inefficiencies in thermal systems [7]. Exergetic analysis has been implemented to detect inefficiencies in thermodynamic systems and to improve energy efficiency or reduce energy consumption in geothermal power plants. Some authors, [8,9,10,11], have specifically focused on the exergetic evaluation of the ORC cycle integrated in flash–binary geothermal power plants to increase their energy efficiency and reduce the magnitude of thermodynamic inefficiencies. However, the exergetic analysis is limited to knowing the costs of the thermodynamic streams of the system and the costs of its irreversible processes. However, in practice, exergetic analysis is used in combination with exergoeconomic analysis. Mohammadkhani et al. [12] describe exergoeconomic analysis as a relatively new and powerful method that combines the second law of thermodynamics with economics. In exergoeconomics, the costs associated with thermodynamic inefficiencies are related to the total product cost of an energy conversion system [13]. Then, in order to reduce the cost of products and improve the energy transformation process in geothermal power plants, the community has conducted some research in this area [14,15,16,17]. 

In addition to exergetic and exergoeconomic methods, exergoenvironmental analysis, which is a combination of exergetic analysis and life cycle assessment (LCA) concepts, is a recent method for evaluating the environmental impacts of energy conversion systems, such as geothermal power plants with waste heat recovery [18]. Several authors have investigated this type of systems from this perspective [19,20,21,22,23,24]. Table 1 shows previous works focused on evaluations of a low temperature ORC cycle. 

It can be seen that none of the previous research has focused on the analysis of the thermodynamic performance of the ORC cycle activated with low-temperature waste heat from flash–binary geothermal power plants from the perspectives of an energetic, exergetic, exergoeconomic and exergoenvironmental performance analysis of its configuration. Therefore, the present study aimsedto fill the existing gap in the literature. In this way, in the literature there are various works related to the analysis of flash–binary power plants, however, it is well known that this type of power plant still wastes a large amount of energy (waste heat) which in a later step is sent to reinjection, largely wasting low temperature geothermal resources. The innovation of this work focuses on the recovery of that energy from the waste heat, proposing the integration of a low-temperature ORC in a flash–binary geothermal power plant and analyzing that recovery to demonstrate the thermodynamic, economic and environmental performance. It is proposed to identify through the 4E evaluation the recovery of residual heat in flash–binary geothermal power plants as an area of opportunity for the use of low-temperature ORCs. The foregoing represents part of the innovation of this work because the recovery of waste heat in binary-flash power geothermal plants through low-temperature ORCs has not yet been evaluated. In this sense, the recovery of waste heat in flash–binary power geothermal plants can be a promising direction that motivates the use of low-temperature ORC because at present, it still does not overcome certain economic barriers to be able to be implemented on a large scale. In this way, the ORC can be incorporated in short periods of time to geothermal power plants already existing or they can also be included in the design of new flash–binary power geothermal plants. In the same way, the comparison of the recovery of waste heat against the direct activation of the ORC cycle through low-temperature geothermal wells is proposed, in order to analyze the advantages and disadvantages of both configurations, as well as their viability in practical applications. The foregoing is part of the novelty of this work as it pursues benefits such as a more efficient use of geothermal resources and a more extensive use of low-temperature ORCs in electricity generation. 

The foregoing represents the motivation of the authors of this work pursuing the primary objective of analyzing the energetic, exergetic, exergoeconomic and exergoenvironmental (4E) performance of an organic Rankine cycle activated with low-temperature waste heat from a flash–binary geothermal power plant. Upon reaching this objective, the aim is: (i) to obtain the energy performance of the cycle, distribution of energy flows in the components and energy efficiencies of the cycle, in order to quantify and identify the components with the greatest energy interactions in the cycle and with the cycle frontiers; (ii) to obtain, through exergetic analysis, the components of the cycle with the greatest thermodynamic inefficiencies, in order to detect the critical components and establish future probabilities of thermodynamic improvements for the components and for the overall cycle in general; (iii) identify, through exergoeconomic analysis, the critical components and streams of the cycle that influence the thermoeconomic performance, in order to detect the cost reduction potential of each stream and system components; (iv) identify, through the exergoenvironmental analysis, the location and magnitude of the components with the greatest environmental impact of the ORC cycle, in order to detect the possible limits of reduction of environmental impact and in the same sense the scope of the exergoenvironmental performance of the cycle.

## 2. Description of the ORC Coupled to the Flash–Binary Geothermal Power Plant

The geothermal power plant is made up of a flash–binary geothermal power plant, and an ORC cycle (binary cycle). This power plant can produce three energy products simultaneously. The flash–binary geothermal power plant is activated with a high-temperature geothermal resource which consists of a liquid–vapor mixture, typical of the existing geothermal resources in the geothermal fields of The Azufres México (T > 250 °C) [25]. The geothermal resource first enters the separator of the single flash geothermal plant, which is separated into two streams, one liquid and one vapor. The steam is directed to the turbine of the simple flash geothermal power plant to generate energy (first product). Once the energy of the steam is harnessed, the steam passes through the condenser to become a liquid and then is reinjected, while the other stream, the liquid hot water, is directed to activate a binary cycle of high temperature (T = 174 °C). Through the evaporator and preheater of the ORC cycle, this hot liquid water comes into indirect contact with an organic fluid, where it transfers its energy. The organic fluid uses the absorbed energy to turn into steam that is used in a turbine to produce energy (second product). Generally, in flash–binary power plants, once the geothermal fluid has given up its energy, it is sent to the reinjection well, wasting a considerable amount of energy still contained in the geothermal fluid. Therefore, the proposal of the present work focuses on the recovery of that energy (waste heat) contained in the waste stream by incorporating a low-temperature ORC cycle; see Figure 1. The purpose is to improve the use of geothermal energy, increase efficiency and obtain additional energy generation (third product), among other economic and environmental benefits. Then, the primary objective is to analyze the energy, exergetic, exergoeconomic and exergoenvironmental performance of an organic Rankine cycle coupled to a low-temperature level of a flash–binary geothermal power plant, in order to recover waste heat. Figure 1 shows the configuration of waste heat recovery in the flash–binary power plant from an ORC cycle activated with a low-temperature source of 110 °C.

The operating conditions of the subsystems (single flash and binary cycle) of the flash–binary geothermal power plant are described below. Assuming the actual conditions of The Azufres Mexico geothermal field, the single flash geothermal power plant operates with a vapor stripping pressure of 8 bar and a condensing pressure of 13 kPa [25]. The geothermal fluid at the wellhead has a temperature of approximately 250 °C and a steam quality in the dominant liquid zone of 0.35 to 0.45. On the other hand, the binary cycle is activated with the residual heat from the geothermal separator. The geothermal resource in the form of saturated liquid activates the binary cycle at a temperature of 174 °C. The geothermal resource exits the binary cycle at a temperature of 110 °C. The binary cycle evaporator takes the organic fluid in saturated liquid conditions. The organic fluid in the evaporator reaches the superheated vapor region. Table 2 shows the T-s diagram of each of the subsystems that make up the flash–binary geothermal power plant.

Table 3 shows a detailed description of the ORC cycle. The heat recovery configuration has been chosen based on the waste heat coming from the simple flash and binary cycle geothermal power plants of the geothermal field from The Azufres, México [25], while for the configuration of components of the ORC and operating conditions of the cycle, real data from the low-temperature ORC cycle of the polygeneration geothermal plant in Ixtlán de los Hervores, Michoacán, México have been considered [26]. The ORC operates with R245fa as the working fluid and has a simple configuration (four elements: evaporator I, turbine II, condenser III and pump IV). Regarding the activation energy of the ORC, the waste heat from the geothermal power plant (hot water) enters the evaporator of the ORC at a temperature of 110 °C and leaves at a temperature of 90 °C (streams, 1–2). Once the geothermal fluid has ceded its energy, it is used in a later step for the reinjection process. On the other hand, the organic fluid, stream (3–6), enters the turbine under saturated steam conditions and leaves the condenser in saturated liquid conditions. In Table 1, it can be seen in detail the diagram of components of the ORC and the temperature-specific entropy (T-s) diagram with the fluids involved in the energy conversion process in the cycle. 

As can be seen in the T-s diagram of Table 3, the beginning of the expansion of R245fa starts in the saturation curve. This indicates that in this work the overheating of the working fluid has not been considered, a thermodynamic process that generally occurs at actual operating conditions in organic working fluids. Cao et al. [27], presented research for the optimal design of an ORC in which the superheated temperature is considered. However, because this work focuses on obtaining an estimate of the 4E performance of the ORC cycle activated with low-temperature residual heat, the design of the ORC cycle has not been described in detail at this stage of research. Tontu et al. [28] have assumed the same consideration about the saturated steam conditions at the turbine inlet when evaluating the exergoeconomic and exergoenvironmental performance of an ORC coupled to a steam power plant. In addition, the design and non-design model of the cycle and of the geothermal power plant, in general, is contemplated in future stages of research. In addition, due to the objective that is pursued related to the energetic, exergetic, exergoeconomic, and exergoenvironmental evaluation of the ORC cycle activated with low-temperature residual heat and with geothermal resources, the degree of overheating does not imply drastic changes to the results with the established approach. Finally, the evaluation proposed in this paper has been established and proposed to estimate the 4E performance of an existing ORC cycle. Due to the above, it does not go into detail about the design of the cycle and the selection of the working fluid. The present work focuses on the 4E evaluation and the integration of the ORC cycle in geothermal power plants, that is, the ORC cycle already exists and is designed to operate with R245fa as a working fluid, and through the performance achieved by the ORC, it is proposed to evaluate the feasibility of integrating this technology with existing geothermal power plants or commercially available equipment. In addition, because it is a cycle that already exists, and its commercial disposition is of a basic Rankine cycle, it is for this reason that additional configurations have not been considered in the analysis, such as regeneration to evaluate the performance of the cycle. Variations to the basic ORC configuration could be contemplated in future works.

### 2.1. General Considerations for Modeling

#### 2.1.1. Considerations for ORC Activation

For the analysis of the ORC cycle, two alternatives have been considered as sources of primary activation of the cycle. The activation alternatives are shown in Table 4. In alternative S1, the activation of the ORC cycle is contemplated through the recovery of waste heat from a flash–binary geothermal power plant while in alternative S2, the direct activation of the ORC cycle has been considered, through thermal energy from the geothermal well. For both alternatives, the same quantity and quality of energy have been considered, that is, the same temperature and geothermal activation mass flow (110 °C y 48.7 kg/s). Under these considerations, the performance in terms of energy, exergy, exergoeconomic and exergoenvironmental feasibility of the ORC can be obtained. It should be noted that under the conditions of alternative S1, the monetary cost and environmental impact of the ORC activation stream is practically zero because it is waste heat; that is, the exergoeconomic and exergoenvironmental costs related to the geothermal well are not considered. However, under alternative S2, it is necessary to consider those costs because the energy is not waste heat from some thermal process. Therefore, it is necessary to include the costs of drilling the geothermal well, which is generally and primarily a function of the depth. In this sense, the depth of the geothermal well that activates the Ixtlán de los Hervores, México polygeneration plant has been assumed [13].

#### 2.1.2. Thermodynamic Considerations

In this paper, some assumptions have been made to obtain the performance of the ORC cycle from different thermodynamic perspectives and based on the different activation energy considerations. The main considerations of the ORC cycle are the following:The system operates under steady-state conditions and the kinetic and potential energy have been neglected;The geothermal fluid of activation has been considered to be hot water;The temperature and pressure of the reference state have been assumed to be 25 °C and 100 kPa, respectively;The minimum temperature difference between the condenser and the evaporator has been considered to be 10 °C [29];The temperature of the cooling water has been assumed to be 26 °C;The isentropic efficiency of the turbine and the pump have been considered to be 75 and 80%, respectively [30,31].

## 3. Thermodynamic, Economic and Environmental Models

The harnessing of waste heat from the flash–binary plant through the ORC cycle has been modeled through the Engineering Equation Solver (EES) software. The EES software provides all the intensive and extensive thermodynamic properties of the ORC, and the streams involved in the use of geothermal heat. The principle of conservation of mass and energy in conjunction with equilibrium equations based on exergy, is used in each part of the ORC cycle. In the analysis, each component is considered as a control volume with inlet and outlet streams, and heat and work interactions are considered [32].

### 3.1. Mass Conservation

The general equation for the conservation of mass used is the following [33,34]:(1)$$\frac{dm_{V.C}}{dt}+\sum_{e}{\dot{m}}_{j}-\sum_{s}{\dot{m}}_{j}=0$$

By applying Equation (1) to the organic Rankine cycle activated with heat recovered from the flash–binary geothermal plant, the mass flow balances can be represented as shown in Table 5.

### 3.2. Energy Modeling

It is possible to determine such a flow energy from the first law of thermodynamics, Equation (2) [33,35]:(2)$$\frac{dE_{V.C,k}}{dt}={\dot{Q}}_{V.C,k}-{\dot{W}}_{V.C,k}+\sum_{e}{\dot{m}}_{j}\cdot {\left(h+\frac{V^{2}}{2}+gZ\right)}_{j}-\sum_{s}{\dot{m}}_{j}\cdot {\left(h+\frac{V^{2}}{2}+gZ\right)}_{j}$$

For the ORC coupled to the geothermal plant, the mass balances and efficiency parameters are presented by component in Table 6. In the case of the evaporator, it is necessary to determine the energy flow between the geothermal fluid and the organic fluid, R245fa, of the cycle, while in the case of the condenser, it is necessary to determine the heat transfer between the organic fluid and the cooling water.

#### Heat Exchanger Equipment

The ORC cycle involves at least two heat exchangers: the evaporator and condenser. The first heat transfer process originates between the cycle activation energy and the organic fluid, and the second heat transfer process occurs between the cooling fluid and the organic cycle working fluid. Therefore, it is necessary to determine the area necessary to transfer energy from one fluid to another. Using the heat transfer capacity, the heat flux in the evaporator and condenser can be defined based on the Logarithmic Mean Temperatures Difference (LMTD) method [36]. Through the LMTD method, it is possible to obtain the heat transfer area of the heat exchanger equipment, using the global heat transfer coefficient (U).

To obtain an estimate of the heat transfer area of the evaporator and condenser, an exhaustive review of the global heat transfer coefficients, obtained and used in various investigations, has been carried out. Kim et al. [37] have indicated that the magnitude of the overall heat transfer coefficient for an evaporator operating with R245fa, in the ORC cycle configuration, ranges between 2.5 and 2.77 kW/m^2^ K, while Capata et al. [38] have indicated that the global heat transfer coefficient for the condenser of an ORC has an approximate value of 2.87 kW/m^2^ K. It is necessary to point out that both in the evaporator and in the condenser there are phase changes in the refrigerant R245fa, so it is necessary to verify that there are no temperature crossovers between the hot stream (for example: in the evaporator stream 1–2) and cold stream (in the evaporator streams 6–3), so that the LMTD method can be applied [39]. In this work, it has been verified that these temperature crossings do not occur between streams in the heat exchangers. Table 7 shows the heat transfer parameters and criteria used in the evaluation of the heat transfer area of the evaporator and condenser of the ORC cycle [40,41].

In order to calculate the heat transfer rate and analyze the behavior it is necessary to implement the number of heat transfer units (ε-NTU). For example, to analyze the evaporator heat transfer units, heat exchanger effectiveness is defined as the actual heat transfer capacity divided by the maximum possible heat transfer. In this case, the heat transfer effectiveness of the evaporator is a function of temperatures, Equation (3) [42]:(3)$$\varepsilon_{I}=\frac{T_{3}-T_{2}}{T_{3}-T_{6}}$$

### 3.3. Exergetic Modeling

Exergetic analysis, based on the second law of thermodynamics, allows us to consider and calculate the irreversibility of a system. Exergy is defined as the maximum work that can be obtained by an energy flow in equilibrium with the reference temperature of the environment. The total exergy rate of a flow is defined by Equation (4) [43]:(4)$$\dot{E}x=\dot{E}x^{PH}+\dot{E}x^{CH}+\dot{E}x^{KN}+\dot{E}x^{PT}$$

In this work, the chemical exergy rate is not considered and kinetic and potential exergy rates are also neglected. Consequently, the physical exergy rate can be expressed as [44]:(5)$${\dot{E}}_{x,j}={\dot{E}}_{x,j}^{PH}={\dot{m}}_{j}\cdot e_{x,j}$$
where the specific exergy can be written as [45]:(6)$$e_{x,j}=\left(h_{j}-h_{0}\right)-T_{0}\cdot \left(s_{j}-s_{0}\right)$$

Regarding the exergy destruction rate for a system in the steady-state process, the exergy rates associated with heat and power can be expressed in steady flow as follows [46]:(7)$$\dot{E}x_{w,j}={\dot{W}}_{j}$$
(8)$${\dot{Ex}}_{q,j}={\dot{Q}}_{j}\cdot \left(1-\frac{T_{0}}{T_{j}}\right)$$

The general equation to obtain the exergy destruction rate of a system, through a general approach, is obtained from the combination of the concepts of the first and second law of thermodynamics, Equation (9) [47]:(9)$$\dot{E}x_{D,k}=\sum {\dot{Q}}_{j,k}\cdot \left(1-\frac{T_{0}}{T_{j}}\right)-{\dot{W}}_{j,k}+\sum_{e}{\dot{m}}_{j}\cdot e_{x,j}-\sum_{s}{\dot{m}}_{j,k}\cdot e_{x,j}$$

The exergy destruction of the components of a system can also be obtained using conventional exergy analysis. This analysis is useful to obtain a more precise thermoeconomic evaluation of the components. However, in the case of components where their resources and products are not defined, it is necessary to use Equation (9) to obtain the exergy destruction rate. Conventional exergy analysis uses the fuel–product terminology for the analysis of thermal systems, such as the case of the ORC cycle [48]. Applying this terminology, the energy available from the resource that activates a component of the system is called the exergy rate of Fuel $\left(\dot{E}x_{F,k}\right)$, and the energy available from the product of a system component is called the exergy rate of product $\left(\dot{E}x_{P,k}\right)$. In the same way, the exergy destruction rate for each component $\left(\dot{E}x_{D,k}\right)$ can be expressed as in Equation (10). The exergy destruction rate of the total system can be represented as in Equation (11). Finally, the exergy efficiency $\left(\varepsilon_{k}\right)$ and exergy destruction ratio of each component $\left(y_{D,tot}^{*}\right)$, can be expressed as in Equations (12) and (13), respectively [49].
(10)$${\dot{Ex}}_{D,k}={\dot{Ex}}_{F,k}-{\dot{Ex}}_{P,k}$$
(11)$${\dot{Ex}}_{F,tot}={\dot{Ex}}_{P,tot}+{\dot{Ex}}_{D,tot}+{\dot{Ex}}_{L,tot}$$
(12)$$\varepsilon_{k}={\left(\frac{{\dot{Ex}}_{P}}{{\dot{Ex}}_{F}}\right)}_{k}$$
(13)$$y_{D,tot}^{*}=\frac{{\dot{Ex}}_{D,k}}{{\dot{Ex}}_{D,tot}}$$

Table 8, shows the exergy destruction rate balances and exergy efficiencies for each component of the ORC cycle and for the total cycle. The total fuel of the ORC cycle is solely a function of the evaporator resource, while the product depends on the resource consumed by the pump and the product of the turbine. As can be seen, the exergetic efficiency of the condenser has not been defined. Zhao y Wang [17], indicate that the condensers serve other components of the system and their resources or exergy products are not defined.

To analyze the total exergy destruction rate of the ORC cycle, it is necessary to determine the exergy loss rate $\left(\dot{E}x_{L,tot}\right)$. Concerning the above, the exergy flow dissipated by the condenser would represent the exergy losses of the system (ORC cycle). However, the exergy loss in the condenser is related to the transfer of thermal energy at ambient temperature. Exergy loss is the transfer of exergy from the system to the surroundings. Taking into account the limits of the analysis of fixed components at ambient temperature, the exergy loss is a very small magnitude and its thermodynamic quality is practically insignificant [50]. However, the total cycle exergy loss rate can be determined by Equation (14):(14)$${\dot{Ex}}_{L,tot}=\dot{E}x_{9}-\dot{E}x_{8}$$

### 3.4. Exergoeconomic Modeling/Economic Feasibility

Exergoeconomics is a powerful and influential knowledge driven by a combination of economic and exergy concepts; it helps researchers better understand systems from an exergy and economic point of view. It makes possible the economic design of power systems that cannot be obtained by standard economic models [13,22]. In exergoeconomic analysis, a cost value is associated with each exergy flow within a system, since exergy is considered the only rational basis for assigning the cost of a thermal system [17,46].

#### 3.4.1. Investment Costs

To carry out the exergoeconomic analysis, it is first necessary to evaluate the investment costs of the components of the cycle $\left(Z_{k}\right)$. In this work, the investment costs of the components of the ORC cycle are considered in terms of their nominal capacity. In the case of the turbine and the pump, the investment cost is evaluated based on its nominal power $\left({\dot{W}}_{k}\right)$, while in the case of the evaporator and the condenser, the investment cost is estimated based on the heat transfer area $\left(A_{k}\right)$ and the type of heat exchanger. On the other hand, it should be noted that the investment costs of the components also depend on factors such as manufacturing processes, materials, instrumentation and control [51]. Table 9 shows the equations to estimate the costs of the components of the ORC cycle. Such cost equations have been obtained through free access literature and by direct quotation with manufacturers [5,46]. Finally, the cost of the working fluid has not been included in the analysis of this work. The influence of the working fluid on the costs of a thermodynamic cycle is mainly related to the heat transfer capacity; as the heat transfer capacity is decreased, this is reflected in an increase in the heat transfer area and consequently in the cost. Wei et al. [52] have carried out research related to the influence of working fluids in organic Rankine cycles; in the work, it is possible to appreciate the variation of investment costs depending on the working fluids used.

In the case of alternative S2, it is necessary to include the cost of the geothermal well. The investment costs of these wells are mainly related to the drilling depth. For geothermal wells of low and moderate temperatures (90–150 °C), the depths range between 200 and 400 m and their drilling cost is around 2150 $/m of drilling [53].

#### 3.4.2. Capital Recovery Factor

The Capital Recovery Factor (CRF) converts a present value into a flow of annual payments during a specific time, it can be estimated from Equation (15). The CRF is a function of the useful life of the components in the cycle (*n*) and the interest rate (*i*). In this paper, an interest rate of 10% and a useful life of the cycle components of 20 years were used [54].
(15)$$CRF=\frac{i\cdot {\left(1+i\right)}^{n}}{{\left(1+i\right)}^{n}-1}$$

#### 3.4.3. Cost Rate

The cost rate $\left({\dot{Z}}_{k}\right)$ represents the investment cost of each component of the ORC cycle as a function of the annual operation time, and is calculated by Equation (16). In this paper, an operating time of 7446 h per year $\left(t_{op}\right)$ and a typical maintenance factor (∅), used for the evaluation of geothermal power plants, of 1.06 have been assumed [44]: (16)$${\dot{Z}}_{k}=\frac{Z_{k}\cdot CRF\cdot \phi }{t_{op}\cdot 3600}$$

#### 3.4.4. Cost Balance

The exergoeconomic analysis provides information on the cost formation process and the unit exergy cost of each stream. This analysis, in a similar way to the exergetic analysis, is carried out by forming cost balance equations and auxiliary equations for each component of the cycle, expressed in the form [55,56]:(17)$$\sum_{s}{\dot{C}}_{j,k}+{\dot{C}}_{w,k}={\dot{C}}_{q,k}+\sum_{e}{\dot{C}}_{j,k}+{\dot{Z}}_{k}$$
(18)$${\dot{C}}_{j}=c_{j}\cdot {\dot{E}}_{x,j}$$

For the exergoeconomic modeling of this system (ORC cycle), the cost balance and the auxiliary equations are considered as presented in Table 10 [57]. In the case of the activation stream of the ORC (alternative S1), the cost per unit of exergy has a value of zero. This is because it is waste heat that has no cost. On the other hand, the cooling fluid, involved in the ORC cycle condenser, is assumed to have the same value [46,58].

#### 3.4.5. Thermoeconomic Evaluation

A thermoeconomic evaluation of the cost variables, implementing the fuel–product terminology of the conventional exergoeconomic analysis, accurately reflects the cost distributions in the components. Thermoeconomic variables are expressed as specific cost of fuel $\left(cF_{k}\right)$, as seen in Equation (19). The specific cost of product $\left(cP_{k}\right)$ can be calculated by Equation (20). The cost rate of exergy destruction $\left({\dot{C}}_{D,k}\right)$ can be calculated by Equation (21). The cost rate of exergy loss $\left({\dot{C}}_{L,tot}\right)$ is expressed in Equation (22). Relative cost difference $\left(r_{k}\right)$ can be calculated by Equation (23), and the exergoeconomic factor $\left(f_{k}\right)$, senn in Equation (24), represent key functions to determine the economic performance of thermal systems from a thermodynamic point of view [59]. The equations involved in a thermoeconomic evaluation, within a conventional exergoeconomic analysis, are presented below:(19)$$cF_{k}=\frac{{\dot{C}}_{F,k}}{\dot{E}x_{F,k}}$$
(20)$$cP_{k}=\frac{{\dot{C}}_{P,k}}{\dot{E}x_{P,k}}$$
(21)$${\dot{C}}_{D,k}=cF_{k}\cdot \dot{E}x_{D,k}$$
(22)$${\dot{C}}_{L,tot}=cF_{tot}\cdot \dot{E}x_{L,tot}$$
(23)$$r_{k}=\frac{cP_{k}-cF_{k}}{cF_{k}}$$
(24)$$f_{k}=\frac{{\dot{Z}}_{k}}{{\dot{Z}}_{k}+\left({\dot{C}}_{D,k}+{\dot{C}}_{L,k}\right)}$$

Economic feasibility differs from exergoeconomic analysis because it uses economic indicators to determine the economic profitability of a system, instead of exergy concepts. An economic feasibility analysis indicates the strengths and weaknesses in terms of economics of a thermal system. However, the application of these two different approaches allows knowing the economic benefits of the system from different perspectives. 

To perform the economic feasibility analysis, it is necessary to obtain the investment or capital costs, in this case from the ORC (Section 3.4.1). For economic feasibility, it is necessary to determine the annualized investment cost; this cost represents an annual outlay. In geothermal power plants, the annualized cost is generally estimated from an interest rate and the useful life of the components [60]. Another disbursement is represented by the costs of operation and maintenance of the ORC cycle. The operation and maintenance costs for geothermal plants activated with low-temperature waste heat can be estimated from the operation time, the cycle power and a unit cost of 0.0012 $/kWh [61]. Finally, the income from the sale of electricity is the only monetary income of the system. Income from the sale of electricity depends on the time of operation, the availability of the ORC cycle (85%), the power generated and the cost of electricity sale of 0.08 $/kWh [62]. 

Once the income and expenditures of the system have been evaluated, the economic indicators are obtained: cash flow (CF), net present value (NPV) and simple return payback (SRP). According to the information presented, a positive annual benefit indicates a possible favorable economic feasibility. However, to be certain of a favorable economic feasibility, an evaluation of the NPV and SRP has to be carried out to guarantee the economic feasibility of the system [60]. Table 11 shows the economic parameters and economic indicators involved in the analysis of economic feasibility. 

### 3.5. Exergoenvironmental Modeling

Energy production systems have significant impacts on the environment; therefore, it would seem essential to analyze these systems in order to reduce losses in an environmentally friendly way [21]. The exergoenvironmental analysis is based on the combination of exergetic analysis and LCA (Life Cycle Assessment) methods [63]. In exergoenvironmental analysis, exergy rates are used to determine environmental impacts.

#### 3.5.1. Environmental Impact

The environmental impact of each component of the cycle is related to the useful life of the component. Taking into account that the impact of energy production systems on the environment is essential, the equations shown in Table 12 allow estimating the environmental impact of the components [21].

The environmental impact is mainly a function of the manufacturing materials and the weight of the components [20]. In this paper, the correlations and values shown in Table 10 have been used. In the same way, the weight of the components has been determined based on their nominal capacity, the power for the equipment that produces or consumes power, and the heat transfer area for heat exchanger devices. Once the weight of the component is determined, the environmental impact of the component is evaluated $\left(Y_{k}\right)$, by Equation (25):(25)$$Y_{k}=b_{m,k}\cdot \omega_{k}$$

#### 3.5.2. Environmental Impact Rate

The environmental impact rate of the component $\left({\dot{Y}}_{k}\right)$, can be determined using ecological indicators, as discussed by Cavalcanti [64]. To evaluate the environmental impact of the system, the Eco-indicator 99 has been considered; its value represents one-thousandth of the annual environmental impact of an average European inhabitant [23,65]. The environmental impact rate is a function of the useful life of the component and the operating time in hours per year of the components [20,21]:(26)$${\dot{Y}}_{K}=\frac{Y_{k}}{3600\cdot t_{op}\cdot n}$$

Finally, the total environmental impact rate includes the environmental impact indices that are defined concerning manufacturing, transportation, installation, operation, maintenance and disposal [44]. In order to obtain an estimate of the environmental impacts, here only the environmental impact of the components associated with manufacturing has been considered [21]. 

The foregoing is important, because in relation to the environmental impact, the scope of the work is limited to estimating the environmental impacts of the components, to later determine, in combination with the application of the exergy analysis, the environmental impacts of the streams of the ORC cycle. Equation (27) defines the total environmental impact rate for a component of a thermal system [24,66]:(27)$${\dot{Y}}_{k}^{Total}={\dot{Y}}_{k}^{CO}+{\dot{Y}}_{k}^{OM}+{\dot{Y}}_{k}^{DI}$$

#### 3.5.3. Exergoenvironmental Balance

The environmental impact balance for each component of the system indicates that the sum of the environmental impacts associated with all the input flows (of the term related to the production of pollutants within the component) added to the term related to the environmental impact of the component is equal to the sum of the environmental impacts associated with all outflows [67]. Equation (28) presents the general equation for the environmental impact balance [68]:(28)$$\sum_{o}{\dot{B}}_{j,k}={\dot{Y}}_{k}+\sum_{i}{\dot{B}}_{j,k}$$
(29)$${\dot{B}}_{j}=b_{j}\cdot {\dot{E}}_{x,j}$$

The exergoenvironmental balance equations and auxiliary equations required for each component of the organic Rankine cycle are listed in Table 13 [44]. In the case of alternative S1, because the ORC cycle is activated with waste heat from a renewable energy source, and because an additional product is being obtained from the waste energy, the activation stream of the ORC cycle does not imply an environmental impact. However, in the case of alternative S2, which involves the drilling of a geothermal well, the activation stream does reach an environmental impact. Described in other words, it is necessary to include the environmental impact of the geothermal well in the exergoenvironmental analysis. Colucci et al. [69] indicate that the environmental impact rate of a geothermal well can amount to 1.13 mpts/s.

#### 3.5.4. Environmental Assessment

The main goal of the environmental evaluation, within an exergoenvironmental analysis, is to reduce the environmental and ecological impacts and for this purpose, the exergoenvironmental variables must be defined. The exergoenvironmental variables that define the environmental impact indices of each component are the specific environmental impact per exergy unit of the fuel $\left(bF_{k}\right)$ and the specific environmental impact per exergy unit of the product $\left(bP_{k}\right)$ [46]. Other relevant variables used to evaluate the environmental impact indices are the rate of the environmental impact of the exergy destroyed $\left({\dot{B}}_{D,k}\right)$, the environmental impact rate of exergy loss $\left({\dot{B}}_{L,tot}\right)$, the relative environmental impact difference $\left(r_{b,k}\right)$ and the exergoenvironmental factor $\left(f_{b,k}\right)$. All the variables for the evaluation of the environmental impact are governed by the definition of the equations shown in Table 14.

## 4. Results and Discussions

In this section, the results and discussions of this paper are presented. The Engineering Equation Solver (EES) software has been used to solve the equations of the organic Rankine cycle, activated with waste heat from a flash–binary geothermal power plant and with thermal energy coming directly from a geothermal well. In this way, each component is considered as a control volume and the laws of thermodynamics have been applied to perform the energetic and exergetic evaluation. Additionally, the exergy concepts have been combined with economic and environmental concepts to obtain the exergoeconomic and exergoenvironmental performance of the cycle coupled with the geothermal power plant.

The thermophysical properties of each thermodynamic state established for the ORC cycle are presented in Table 15.

From the evaluation of the thermophysical properties of the thermodynamic states of the ORC cycle, it is possible to apply the thermodynamic models to determine the energy, exergetic, exergoeconomic and exergoenvironmental benefits of the cycle. On the other hand, as the temperature of the activation resource of a thermodynamic system increases, the performance of the thermal system increases. However, it is necessary to emphasize that when it comes to observing this behavior, it is necessary to verify if there are no temperature crossovers in the heat exchanger equipment, to guarantee that the model and evaluation are carried out correctly. In this sense, Figure 2 shows a variation of the temperature of the geothermal resource against the discharge stream of R245fa from the evaporator. In Figure 2, it can also be observed the behavior of the pressure in the evaporator against the activation temperature of the ORC cycle. It should be noted that the pressure values can be increased from 1200–2000 kPa by varying the temperature of the geothermal resource from 110 to 135 °C. On the other hand, the closest approach between the streams that are involved in the activation of the ORC cycle does not prevail when the temperature of the geothermal resource increases; rather, it originates when the phase change of R245fa begins [31].

### 4.1. Energy Analysis Results

The results of the energy analysis for both alternatives (S1 and S2) are shown in Table 16. Under the established conditions of heat recovery, the evaporator uses 4110 kW, of which 477.9 kW of power can be obtained in the turbine. Subtracting the power consumed by the pump (19.44 kW), the cycle manages to convert 458.4 kW of power; this power represents the net product of the thermodynamic cycle. In the same Table 16, it can be seen that, when comparing the energy inputs and outputs of the cycle, the condenser dissipates 88.85% of the activation energy of the cycle to the surroundings (energy in the evaporator/waste heat used), while only 11.15% is converted into a useful product (net cycle power). This allows for identifying the energy distribution of the components of the cycle. In the same way, the condenser could represent great opportunities for thermodynamic improvements of the ORC cycle, since it wastes a large amount of energy. Meanwhile, the turbine represents the opportunity to improve the conversion of thermal energy to electricity.

Table 17, shows the results obtained for the heat transfer parameters. The equipment that implies a bigger heat transfer area in the organic Rankine cycle is the condenser with 93.86 m^2^ necessary to dissipate the 3654 kW of the Cycle. The evaporator requires 62.72 m^2^ of heat transfer area for the R245fa to recover the energy contained in the waste heat or geothermal resource and to obtain 18.09 kg/s of saturated steam at a temperature of 100 °C and a pressure of 1269 kPa in the turbine inlet. Another notorious impact when reducing LMTD is the increase in heat transfer effectiveness, which can be seen in the condenser in Table 17. In the case of the evaporator, the heat transfer effectiveness represents a value of 86.56%. It is not possible to decrease the LMTD to a value close to zero because it would imply obtaining an infinite heat transfer area. 

Once the design parameters have been obtained, such as the heat transfer area of the heat exchanger equipment (evaporator and condenser), the behavior of the ORC cycle can be analyzed based on the determined design parameters. Figure 3 shows the variation of the turbine power, the pump power, the net power of the ORC cycle and the efficiency. In Figure 3, it can be seen that as the activation temperature of the ORC cycle increases, the power output of the turbine increases, and consequently the net power output of the cycle also increases. In the same way, the efficiency and the power consumed by the pump are increased. However, it should be noted that as the activation temperature of the cycle increases, the pump represents a higher energy consumption. Therefore, at lower temperatures, the power difference between the turbine and the net power of the cycle is lower, as can be seen in Figure 3. Finally, the efficiency represents an increase in efficiency of 2% by increasing the activation temperature 15 °C.

### 4.2. Exergy Analysis Results

Figure 4 shows the results of the exergy analysis obtained for alternatives S1 and S2. In the image, the exergy flow rate of the ORC cycle and the exergy efficiencies of the evaporator, turbine and pump can be seen. In terms of exergy rate, it can be seen that in the geothermal fluid stream (1–2), which activates the ORC cycle, thermodynamic state 1 is the one that contains the largest amount of available energy. However, it is also seen that the exergy rate of thermodynamic state 2 ranks second in terms of the amount of energy available. It is important to highlight that the exergy contained in thermodynamic state 2 is intended for reinjection, that is, it is waste heat from the plant, even with a good quality of energy, which is wasted. Therefore, the energy still contained in this thermodynamic state represents opportunities to be used in some other thermal process and even possibly in the generation of electricity again. Regarding the exergy rate flow of the ORC cycle, corresponding to the working fluid R245fa, the exergy rate of state 3 is the highest in the cycle, followed by the power exergy obtained in the turbine. Finally, the exergy rate flow of the cooling water streams (8–9) is practically null because the cycle dissipates the energy of the condenser at a temperature very close to the reference temperature (T_0_) used. Figure 4 also shows the exergetic efficiencies of the components of the ORC cycle, except for the condenser. This is because the products and resources of this device are not defined, and it only serves as a means for the rest of the components to achieve the power product of the ORC cycle. Seen from another point of view, in the condenser, the conversion of R245fa vapor to liquid is achieved through the dissipation of energy, which represents a loss of exergy but not a product. On the other hand, it can be seen that the component with the highest exergetic efficiency is the evaporator, followed by the turbine and finally the pump. All the aforementioned components have exergetic efficiencies greater than 70%, while the ORC cycle reaches a global exergetic efficiency of 55.34%.

Figure 5 shows the exergy destruction rate of the components of the ORC cycle and the ratios of exergy destruction of each component against the total exergy destruction of the cycle for alternatives S1 and S2. In this way, the results of the exergy destruction analysis reflect that the evaporator is the component with the highest exergy destruction rate (128.9 kW), followed by the turbine (110.4 kW), the condenser (66.11 kW) and finally the pump. (4.69 kW). Based on these results, it can be verified that although the condenser dissipates a large amount of energy to the atmosphere (3654 kW), this energy is not of good quality and it is essential to dissipate it for the cycle to operate correctly under the thermal machine concept. In addition, this energy is dissipated at a temperature very close to ambient (T_0_), so the exergy destruction rate is lower than in the previous components (evaporator and turbine). Regarding the results found, the analysis has turned out to be relevant to identifying the components with the greatest magnitude of exergy destruction rate: the evaporator and the turbine. In this way, once these components have been identified, the objective for future work can be to detect opportunities to improve their thermodynamic performance; that is, to focus the analysis only on the evaporator and turbine to reduce their rate of destruction exergy. On the other hand, the exergy destruction ratio has made it possible to identify the percentage of exergy destruction of each component based on the total exergy destruction of the ORC cycle (310.1 kW). In this aspect, the evaporator and the turbine represent 41.26 and 35.6% of the total destruction of the ORC cycle, respectively. Finally, in Figure 5, the exergy rates of the fuel and the products of each component can also be seen. Notably, it can be seen that both the exergy rate of the fuel and the product are higher in the evaporator, followed by those of the turbine and finally those of the pump. It is important to note that this is because the components are interconnected and in the ORC cycle configuration part of the product of one component represents the fuel of the subsequent component. This makes it possible to propose future work to determine the thermodynamic inefficiencies of the components due to the interaction between them [70]. On the other hand, as mentioned above, it is not possible to define the exergy of the fuel and the product for the condenser, however, if it were possible to define them, they would have a higher exergy than the pump because they are at a higher hierarchical level in the energetic interaction in the cycle. This can be seen in the exergy destruction of the condenser, which is higher than that of the pump. If an analogy is made with everyday life, the exergy destruction rate of the cycle components can be seen in Figure 5, as a descending magnitude scale in (kW) starting at the evaporator and ending at the pump. 

Figure 6 shows the behavior of the exergy destruction rate in the components and the exergetic efficiency of the ORC cycle as a function of the activation energy temperature. It can also be observed, in a temperature range from 110 to 135 °C, that the highest exergy destruction rate of the ORC cycle always corresponds to the evaporator followed by the exergy destruction rate in the turbine in the entire temperature range; the corresponding values for the condenser and the pump are in third and fourth place, respectively. As for the difference between the exergy destruction rate of the turbine and the evaporator, Figure 6 shows that it decreases as the temperature of the resource increases. As can be seen in Figure 6, all components increase exergy destruction rate as temperature rises. The evaporator shows the largest of these increases, going from 130 kW for 110 °C and reaching an exergy destruction rate of 300 kW for 135 °C. With this, a 15 °C increase in temperature of the geothermal resource can cause the exergy destruction rate to double at least in all components. On the other hand, in Figure 6, the behavior of the exergetic efficiency of the ORC cycle can also be seen with respect to the temperature of the activation resource; as the temperature of the resource increases, this efficiency increases by the second law. However, unlike the exergy destruction, for the efficiency, the increase is not so remarkable, and the same temperature increase in the activation resource of 15 °C, thisrepresents an increase in the exergy efficiency of less than 3%.

Figure 7 shows the total exergy rate flows of the ORC cycle. The total exergy rate of the fuel that activates the ORC cycle has a magnitude of 828.4 kW, that is, this is the value of the maximum energy available for the production of the products. Therefore, the total product of the cycle is represented by the net power of the thermodynamic cycle, equivalent to 458.4 kW. However, in the course of the transformation from fuel to product, due to the internal irreversibilities of the ORC cycle, 310.1 kW in total exergy is destroyed. In a collateral direction, in this same energy transformation process, the cycle requires a necessary exergy loss of 59.91 kW. According to these results, the exergy destruction rate could be avoided by improving the design of the components and the interaction between them, which implies goals for future work [71]. It can be seen that the total exergy destruction rate of the cycle represents 37.4% of the exergy rate of the fuel and 67.64% of the exergy rate of the product. On the other hand, it is not possible to avoid the destruction of exergy since the ORC cycle, similar to a thermal machine, requires energy to be dissipated for its operation.

### 4.3. Results of the Exergoeconomic Analysis and Economic Feasibility

In this subsection, the results obtained from the exergoeconomic analysis and economic feasibility of the ORC cycle activated with waste heat from the flash–binary geothermal power plant (alternative S1) and with geothermal energy obtained directly through a geothermal well (alternative S2) are presented.

#### 4.3.1. Results of the Investment Cost Estimation

Table 18 shows the results of the estimation of the investment costs of the components of the ORC cycle and the cost proportions that each component represents with respect to the total cost of the ORC cycle. Both for alternative S1 and for alternative S2, the investment costs of the ORC cycle amount to a total amount of 775,678 USD. This amount is distributed through its components in the following proportions: in the first place is the turbine with a percentage of the total cost of 58.08%, in second place is the condenser with 24.2%, in third place is the evaporator with 16.17% and in fourth place is the pump with only 1.55 %. In the same way, the above can be seen in Table 18, where the turbine and the condenser stand out from the rest of the components. Both the turbine and the condenser involve thermodynamic parameters that are the main factors that lead to high investment costs (with respect to the rest of the components), for example, higher nominal capacities. Through these results, it has been possible to detect that it is not possible to reduce the cost of the turbine once its nominal capacity is already established. However, it is possible to improve its thermodynamic performance by improving parameters such as efficiency. The foregoing will not reduce the investment cost, because the nominal output capacity of the turbine for which it has been designed is fixed, however, it is possible to improve its performance in the use of the energy contained in the R245fa so to convert it more effectively to power. Therefore, the benefit will be reflected in the economic profitability of the cycle. On the other hand, as mentioned above, the condenser is the other component of the cycle that involves a high cost. This cost is mainly due to a design factor, the LMTD (13.56 °C); as the LMTD decreases, the heat transfer area increases and consequently the investment cost. However, a small LMTD implies high effectiveness in transferring units of thermal energy; accordingly, to achieve a reduction in the cost of the condenser, the effectiveness would have to be decreased, which may or may not be very convenient. The foregoing motivates future work to determine the best cost–benefit ratio based on parameters such as the heat transfer area and effectiveness, for example for the condenser.

On the other hand, in Table 18, it can also be seen the great difference involved in activating the ORC cycle with waste heat (S1) and direct thermal energy from the geothermal well (S2). This difference is reflected in the drilling costs of the geothermal well, amounting to 1,258,177.72 USD. This amount represents practically twice the investment cost of the ORC cycle, so the economic profitability of operating the ORC cycle with the form of activation of alternative S2 becomes unattractive.

#### 4.3.2. Results of the Exergoeconomic Analysis

Figure 8 shows the results of the exergoeconomic analysis for the specific cost per unit of exergy of the streams of the ORC cycle. Figure 8 presents a simplified comparison of the specific costs per unit of exergy ($/GJ) of the streams and thermodynamic states of the ORC cycle evaluated under the two activation alternatives (S1 and S2). In this way, it can be seen that the specific costs per unit of exergy of the thermodynamic states of the ORC cycle operating under the conditions of alternative S2 exceed the same costs corresponding to alternative S1. This is because the evaluation conditions of alternative S2 involve the cost of drilling the geothermal well, and due to this, the specific cost per unit of exergy is reflected in all the thermodynamic states of the ORC cycle. That is, when considering the cost of the geothermal well, the specific cost per unit of exergy of the activation energy is different from zero, so the costs are increased compared to alternative S1. Figure 8 shows that after the analysis, the specific costs per exergy unit of alternative S2 are 25% above the costs corresponding to alternative S1. Among the highest costs of the ORC cycle are the specific cost per unit of exergy corresponding to power generation, the exergetic cost of the condensed steam of R245fa and the specific cost per unit of exergy of the state of feed of R245fa to the evaporator. The same behavior of the specific costs per unit of exergy is maintained for both alternatives; they only differ in their magnitude, the costs of alternative S2 being higher because for S1 the waste heat is taken at zero cost. However, even though the waste heat from alternative S1 has no cost, the cycle streams do have a cost, although it is less than alternative S2.

Figure 9 shows the results for the rate of the costs ($/h), for the thermodynamic states of the ORC cycle, evaluated under the activation conditions of alternatives S1 and S2. For both alternatives the highest cost rate is the one associated with the power of the turbine then the one corresponding to the saturated steam of R245fa in the discharge of the evaporator followed by the cost rate associated with the geothermal resource in the discharge of the producing well. It can be corroborated that the power generation costs using thermal machines tend to be the highest in a thermodynamic cycle, in this case, the ORC cycle. Also, Figure 9 shows that even when both alternatives have the same order in the cost for each component, the costs for the alternative S2 are higher than the corresponding for the alternative S1 and therefore, in practice, the recovery of low-temperature waste heat to activate an ORC cycle becomes more attractive, due to the lower generation costs. Another important aspect to highlight is that while the highest specific costs per exergy unit of the cycle are the condensed vapor and the compressed liquid of R245fa (see Figure 8), for the rate of the costs, the highest value is the one associated with power production, and in general, and in practice, the cost of the product of the ORC cycle associated with power is the one that has the greatest relevance.

#### 4.3.3. Results for the Thermoeconomic Evaluation

Table 19 shows the results obtained through the thermoeconomic evaluation of the ORC cycle for alternatives S1 and S2. The table shows the values for (i) the specific cost of exergy fuel, (ii) the specific cost of exergy product, (iii) the exergy destruction cost rate, (iv) the relative cost difference and (v) the exergoeconomic factor, for all components of the ORC cycle. It can be seen that the specific cost of exergy fuel of the evaporator of alternative S1 is zero since for this alternative the cost of the evaporator activation stream is waste heat and therefore does not represent a monetary cost or specific cost per unit of exergy. On the other hand, in Table 19, it can be seen that the highest specific cost per unit of exergy is assumed by the pump, since it is activated with the power of the turbine, and the specific cost per unit of exergy associated with the power of the turbine is one of the highest in the thermodynamic cycle. Consequently, the specific cost per exergy unit of the pump is also the highest; this can be seen in Figure 8. However, although the specific cost per unit of exergy of the product of the pump is the highest, this does not imply that the pump has the highest cost rate of the ORC cycle; such a cost rate corresponds to the power generated by the turbine, as shown in Figure 9. Furthermore, when this component operates under alternative S2, it also has the highest cost rate related to irreversibilities, that is, the highest cost due to exergy destruction rate. Although it should be noted that the costs of S2 greatly exceed the specific costs per unit of exergy of S1. Regarding the difference in relative cost for the evaporator with alternative S1, Table 19 shows that this cost tends to infinity; this is because the resource cost of the evaporator is zero. For the same reason, the exergoeconomic factor tends to a value of 100% since the exergy destruction cost of this component is zero, and therefore this factor is only a function of the investment cost rate. Finally, the thermoeconomic viability of the condenser has not been possible to evaluate because its resources and products are not defined. The condenser is a thermal device that is used to complete the closed cycle work in the ORC; it only helps the rest of the components to achieve the objective of converting thermal energy into a useful product such as power. However, it is possible to assess the cost rate of exergy loss by considering the entire ORC cycle as a system. These exergy losses that occur in the condenser reach a higher cost rate for alternative S2.

#### 4.3.4. Economic Feasibility Results

Figure 10 shows the results of the economic feasibility of the ORC cycle activated with the operating conditions of alternative S1. Under these activation conditions, by taking advantage of the waste heat of the flash–binary geothermal power plant, the ORC cycle has shown acceptable economic feasibility as a primary technology for power generation. Through this alternative S1, the cycle can reach an income from the sale of electricity of up to 284,671 USD/year, which consequently results in good behavior for the economic indicators. The annual benefit of the ORC cycle amounts to a value of 189,291 USD, which, due to being being high, causes the economic indicators that represent the feasibility of a system to be motivating. The first indicator, the NPV, reaches a high value (835,860 USD), which translates into a favorable economic return. On the other hand, according to the analysis, the simple return payback is 4.09 years. The foregoing reflects acceptable economic profitability, which represents an area of opportunity for the implementation of ORC cycles in heat recovery in flash–binary geothermal power plants.

Figure 11 shows the results of the estimation of the economic feasibility for the ORC cycle operating with geothermal heat obtained through a producing well or alternative S2. The main difference with respect to alternative S1 is the cost that is directly reflected in the following: the annualized cost, total investment, cash flow, and economic indicators. Although the cash flow of the system is a positive value (100,902 USD), in reality it does not guarantee a favorable economic feasibility. A low cash flow value results in a negative NPV (−669,140 USD). A negative NPV translates into an unfavorable simple return payback that is confirmed by the payback period of 15.15 years for alternative S2. The evaluation of this alternative S2 makes it clear that in order to achieve a favorable economic feasibility in the generation of electricity through low-temperature ORC cycles activated with geothermal wells, it is necessary to find other more efficient alternatives that improve the behavior of this feasibility. For example, these alternatives may include the integration of technologies for the direct use of geothermal energy, to obtain more products from the same geothermal resource, and thus increases the cash flow and consequently the economic indicators. This implies new challenges for future research.

Due to the thermodynamic quality of low-temperature geothermal resources, their profitability is economically unattractive. However, as shown in Figure 12, an increase in the temperature of these geothermal resources would make it profitable to implement low-temperature electricity generation technologies in them. This increase in the temperature of the activation resource of the ORC cycle makes it possible for alternative S2 to be profitable in the generation of energy from ORC technology. However, this increase in temperature implies an increase in the nominal capacity of the ORC cycle and therefore an increase in investment costs. Then, as shown in Figure 12, for the simple return payback to become attractive, reaching desirable values in the economic indicators and achieving an investment payback period of fewer than 5 years, it is necessary to increase the activation temperature of the cycle up to 135 °C. On the other hand, it would be necessary to increase the power of the ORC cycle up to 1217 kW, which means a rise in the nominal capacity up to 2.65 times; this in turn translates into a 51% increase in investment. As the temperature of the geothermal resource increases, the cash flow and the NPV do so as well, and the simple return payback for the investment is reduced, which makes power generation using the S2 alternative attractive. It should be noted that although alternative S1 is economically profitable, this does not indicate that it can considerably improve the profitability of the flash–binary geothermal power plant, because this depends on the quantity and quality of the waste heat that the geothermal plant dissipates to activate the alternative S1.

### 4.4. Results of the Exergoenvironmental Analysis

In this Subsection, the results of the exergoenvironmental analysis of the ORC cycle are presented. Figure 13 shows the results for the weight and environmental impact of the components. The component with the greatest weight in cycle tons is represented by the evaporator with 36.37 tons followed closely by the condenser with 33.82 tons; these two components alone constitute 96% of the total cycle weight of 72.89 tons. On the other hand, although the turbine only represents 3.7% of the total weight, its contribution to the environmental impact is the greatest, and with 1846 Pts, this component alone represents almost 50% of the total points of the 3818 Pts cycle. This is due to the great diversity of the materials and methods with which the turbines are manufactured. However, the environmental impact weight ratio prevails in heat exchange equipment such as the evaporator and the condenser; that is, the evaporator achieves a greater environmental impact than the condenser. On the other hand, the pump represents a very small percentage contribution compared to the rest of the components, both for weight and environmental impact. What is described above is the result of the analysis of each component for the ORC cycle with the alternatives S1 and S2. However, for alternative S2, it is necessary to additionally include the environmental impact of the geothermal well, which represents another environmental impact of great magnitude. Finally, in Figure 13, the total environmental impact of the ORC cycle can also be appreciated. 

Figure 14 shows the environmental impacts per exergy unit of the ORC cycle for alternatives S1 and S2. Figure 14 highlights the great difference between the environmental impacts of alternatives S1 and S2. For the first of these, where the ORC cycle is activated with waste heat, the environmental impacts are very low, practically null compared to the environmental impacts of alternative S2. Regarding alternative S2, the greatest environmental impact per exergy unit is observed at the condenser outlet, that is, the condensed vapor of R245fa. This is because the condenser is the component that dissipates the thermal energy of the thermodynamic cycle, which represents an environmental impact. The discharge of the pump also implies an environmental impact per unit of exergy of great magnitude within the thermodynamic states of the ORC cycle. This impact is essentially related to the power consumption of this component. Another environmental impact per unit of exergy of significant magnitude is that associated with power generation. Finally, to a lesser extent, it is followed by the environmental impact of the steam that enters and leaves the turbine and the environmental impact per exergy unit of the cycle activation geothermal resource. In general, the environmental impacts per exergy unit of S1 represent around 1% of alternative S2. 

Figure 15 shows the environmental impact rates (in mpts/h) of the thermodynamic states of the ORC cycle operating under the two alternatives (S1 and S2). It can be seen in a similar way to Figure 14 that the environmental impacts of the ORC cycle of alternative S1 compared to alternative S2 are reduced regarding alternative S2; the highest environmental impact rates are reflected in the stream of the geothermal resource, while in the closed circuit of the ORC, the highest rates of environmental impact are observed in the saturated steam of R245fa at the outlet of the evaporator and in the power of the turbine. The high environmental impact rates of the geothermal fluid stream are directly related to the environmental impact rate of the geothermal well. The foregoing is also observed in the ORC cycle, such is the case of the environmental impact rate at the discharge of the evaporator; that is, the environmental impact rate of the geothermal well is observed to be reflected mainly in the saturated steam since the geothermal stream is the resource in charge of obtaining that saturated steam. In the same way, the saturated steam that enters the turbine, and consequently the turbine power, also has a high environmental impact rate. Additionally, in a thermodynamic cycle, the product of the cycle will always have a considerable magnitude of the environmental impact rate because all the configurations and uses of the resource have the purpose of obtaining a product, and if the ORC cycle is observed as a system total, the total environmental impact rates would be related only to the power production, that is, to the product of the cycle.

On the other hand, Table 20 shows the results obtained for alternatives S1 and S2 from the analysis of the exergoenvironmental evaluation. In Table 20 it can be seen that all the exergoenvironmental parameters obtained for the ORC cycle operating under the conditions of alternative S1, are lower than the exergoenvironmental parameters obtained with alternative S2. In this way, the greatest environmental impact of exergy destruction rate occurs in the turbine under alternative S2 and the pump has a greater impact per unit of exergy due to the environmental impact related to power generation. Regarding alternative S1, the specific impact per unit of exergy of the fuel is null because it is activated with waste heat; in addition, the impact of this fuel is reflected in the environmental impact is null also of the destruction of exergy rate in the evaporator. Consequently, the difference in relative environmental impact results in an infinite trend. This behavior is also observed in the 100% environmental impact factor of the evaporator, since such a factor is only a function of its environmental impact rate. Finally, the condenser, similar to the exergoeconomic analysis, has not been evaluated because its resources and products are not clearly defined. In this part, it has only been possible to detect the environmental impact of exergy loss, which is essential to know in a condenser because it is the component that interacts with the surroundings through heat dissipation.

The relative environmental impact difference expresses the potential for reducing the environmental impact on each component of the system. In this way, the turbine operating under alternative S1 represents the greatest reduction potential. On the other hand, a low value of the exergoenvironmental factor indicates that the environmental impact rate associated with exergy destruction is dominant compared to the environmental impact of the component. The analysis shows that the values of the exergoenvironmental factor of the turbine, the evaporator and the pump operating under alternative S2 are very low in relation to the same factors for alternative S1.

According to the results obtained, the ORC activated with low-temperature residual heat has great advantages over the ORC cycle activated directly by heat from a geothermal well. The ORC activated with low-temperature waste heat from a flash–binary geothermal power plant has resulted in being highly profitable from the economic and environmental points of view; therefore, the practical application of this setup can be highly recommended. In addition, the ORC activated with residual heat from a flash–binary geothermal power plant represents a very interesting option to be incorporated into existing geothermal power plants that have low-temperature residual heat available; also it is worth considering the part of waste heat recovery in the design of new geothermal power plants. The previous research has resulted from the advantages of using heat recovery instead the direct use of geothermal steam. In this document, new forms of use of geothermal residual heat that had not been previously evaluated have been evaluated. Even though the ORC has been studied for a long time, the cycle had not been evaluated from the perspective of activation with waste heat from a flash–binary power geothermal plant. Among the main and notorious advantages of the proposed ORC configuration, lower cost rates associated with the products of the cycle and the thermodynamic inefficiencies of the cycle have been obtained. Another significant advantage turned out to be a low environmental impact. In relation to the practical advantages, there is a very important area of opportunity to improve the use of low-temperature resources through the application and integration of ORC in flash–binary geothermal power plants. In this direction, there is a market sector available where low-temperature ORC technology on a small-scale can be widely developed. Finally, the use of low-temperature ORC technologies, which still present economic limitations, becomes economically profitable. By implementing the ORC in the recovery of waste heat, the high cost of drilling new geothermal wells is also avoided, this contributes to the maximum use of the geothermal resources already available to be exploited.

## 5. Conclusions

In this paper, the analysis of the energetic, exergetic, exergoeconomic and exergoenvironmental (4E) assessment of an organic Rankine cycle activated by low-temperature waste heat from a flash–binary geothermal power plant and geothermal thermal energy coming directly from a low-temperature geothermal well was presented. The models of the ORC cycle with an activation temperature of 110 °C were proposed. Subsequently, the results to observe the thermodynamic performance of the cycle in combination with economic and environmental concepts were analyzed. The following conclusions were obtained in aspects of the energetic, exergoeconomic and exergoenvironmental analysis:Through the energy analysis, it has been possible to obtain the energy flows of the cycle, the energy efficiency and the nominal capacities of the components of the cycle. In this way, it was possible to identify that 88.85% of the activation energy of the cycle is dissipated in the condenser and only 11.15% of said energy is used or converted into a useful product. On the other hand, based on the energy involved in the exchange equipment and in the equipment that interacts with work on its borders, it has been possible to establish the nominal capacities of the components of the cycle. It was found that the condenser involves 33.17% more heat transfer area than the evaporator to dissipate the cycle energy. The foregoing could suggest that the condenser wastes a large amount of energy, however, this dissipation is necessary for the correct operation of the ORC cycle without mentioning the concept of thermal machine operation. Finally, the energy analysis evidenced the performance by the first law of thermodynamics for the ORC cycle activated with waste heat or heat from a low-temperature geothermal well. In this direction, the ORC cycle has achieved a thermodynamic performance with a thermal efficiency of 11.15% by the first law;Through the exergetic analysis, the thermodynamic performance of the ORC cycle operating under the different activation alternatives was obtained, implementing the criteria of the second law of thermodynamics. Under this approach, it was obtained that the ORC cycle reaches an exergetic efficiency of 55.34%. In the same way, through the exergetic analysis it has been possible to observe that the component with the greatest irreversibilities within the ORC cycle is the evaporator and not the condenser, a component that under the first law has one of the greatest energy wastes. However, the analysis by the second law has allowed detecting the component with greater irreversibilities. Also, by means of exergy destruction analysis, both the total and component irreversibilities have been detected and quantified. In relation to the above, the magnitude of the exergy destruction rate found for each component sets new goals for future research, focused on minimizing this destruction of exergy rate and, consequently, minimizing the total thermodynamic inefficiencies of the thermodynamic cycle in general. Finally, the exergy loss rate in the condenser represents a very small magnitude compared to the total exergy destruction rate. It is important to note that exergy loss rate is unavoidable because the ORC cycle requires energy dissipation, while the exergy destruction rate can be reduced with better design. In this aspect, the cycle destroys 37.4% of the exergy of the fuel and 67.64% of the exergy of the product, which implies high possibilities of improving the thermodynamic performance by the second law of the ORC cycle activated with low-temperature waste heat from a geothermal flash–binary power plant;To carry out the exergoeconomic analysis, it was necessary to first obtain the investment costs of the components. Therefore, it was possible to identify the cost distribution of the ORC cycle for the case of alternative S1, and the cost distribution for the case of alternative S2 that involves a low-temperature geothermal well. For both alternatives, the turbine represents the highest investment cost of the cycle. However, by including the drilling of the geothermal well in alternative S2, the investment cost increases considerably since the cost of the geothermal well is equivalent to another ORC cycle of the same dimensions. This is observed to directly affect the economic profitability, where the S2 is not attractive. While alternative S1 becomes highly profitable at a temperature of 110 °C, alternative S2 would require a larger cycle and a higher activation temperature to become profitable. In this way, the recovery of waste heat in flash–binary geothermal power plants, to activate ORC cycles of small capacity and with low activation temperatures, is attractive. In addition, this implementation of waste heat recovery has payback periods of less than 5 years for the initial investment. On the other hand, through the exergoeconomic analysis, it has been possible to detect the great advantages of alternative S1 over alternative S2. Increases in the cost rate of up to 39.28% were observed for the alternative that implements the use of direct heat from a geothermal well, compared to the waste heat configuration of a flash–binary geothermal power plant. Additionally, the relative cost difference indicates a great potential for cost reduction in the turbine and evaporator of the alternative activation of the ORC cycle by waste heat of the flash–binary geothermal power plant;Finally, the exergoenvironmental analysis reflects the benefits of the ORC cycle analyzed for two alternatives from a perspective that involves exergetic and environmental concepts. Through the exergoenvironmental analysis, it has been possible to identify the environmental impacts of all the components of the cycle, as well as the environmental impact of all the streams of the ORC cycle. In this aspect, for the cycle operating under conditions of alternative S1, the stream with the greatest environmental impact of the cycle is the one associated with power generation, and under the cycle operating under conditions of alternative S2, it is the stream from the geothermal resource. Finally, the relative environmental impact difference and the environmental exergy factor have resulted in great potential to reduce the environmental impacts of the ORC cycle operating under both alternatives. However, the ORC cycle alternative operating in the waste heat recovery configuration of a flash–binary geothermal power plant has resulted in higher exergoeconomic and exergoevironmental performance.

## Figures and Tables

**Figure 1 entropy-24-01832-f001:**
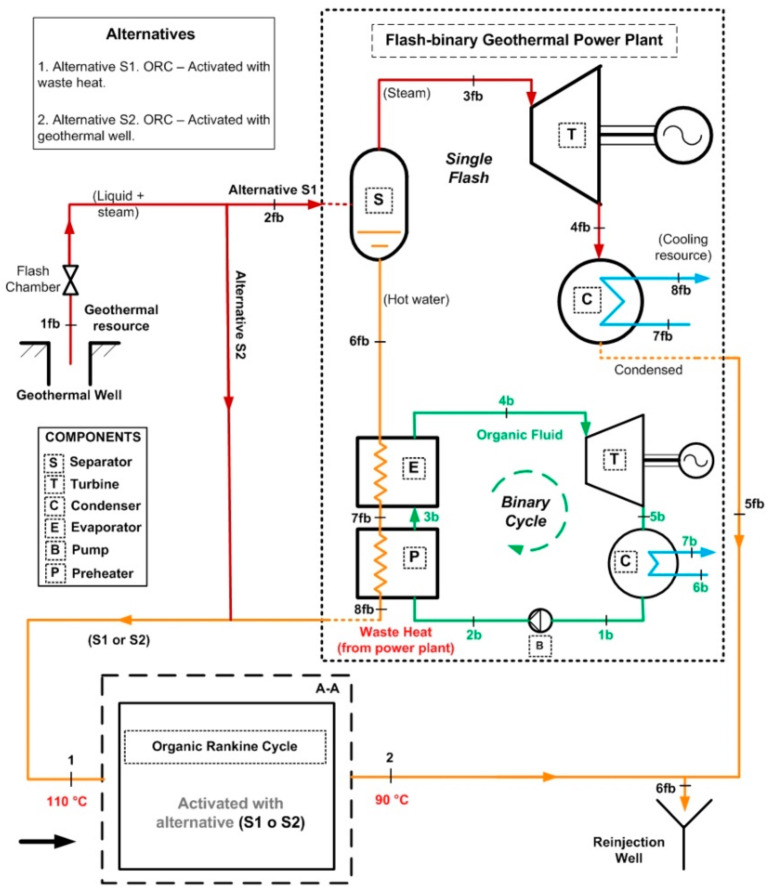
Heat waste recovery by ORC in flash–binary geothermal power plant.

**Figure 2 entropy-24-01832-f002:**
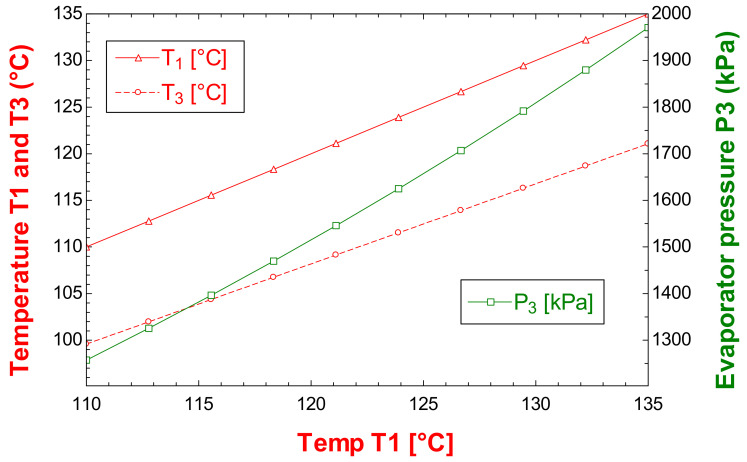
Evaporator temperature as a function of waste heat temperature.

**Figure 3 entropy-24-01832-f003:**
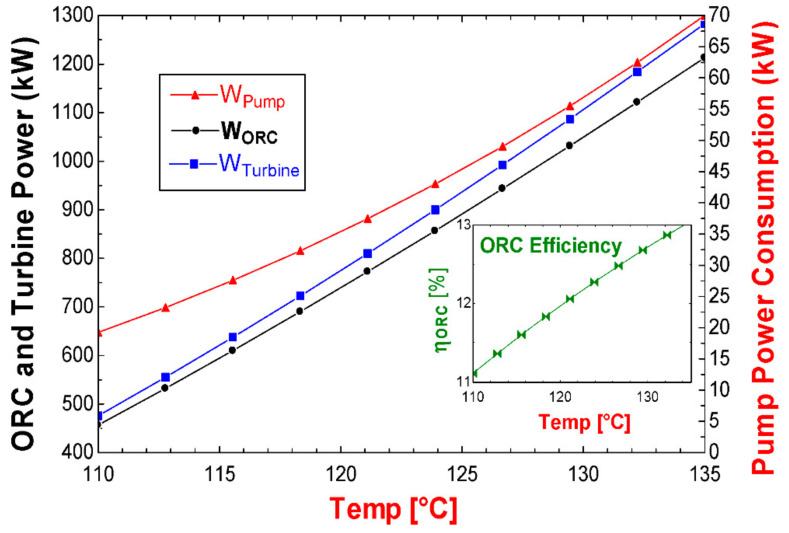
Power behavior and energy efficiency of the ORC cycle.

**Figure 4 entropy-24-01832-f004:**
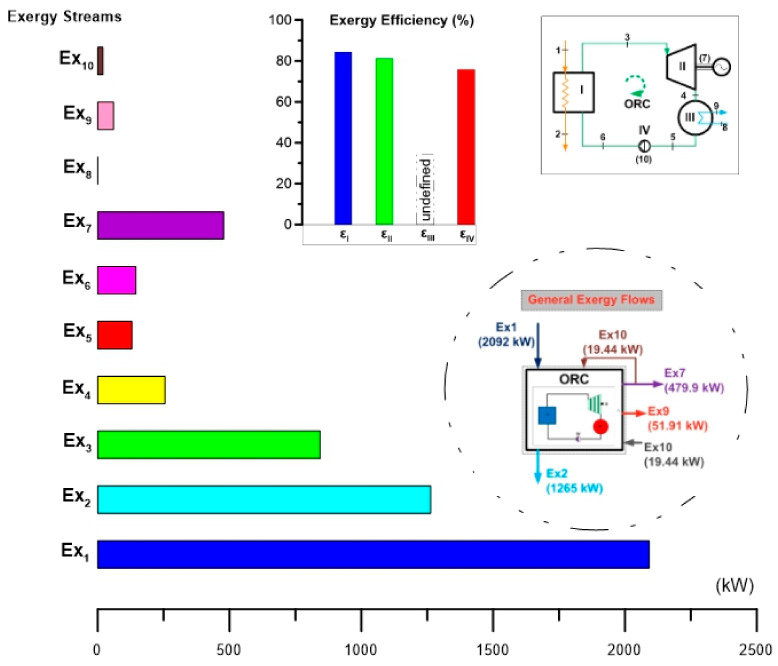
Exergy rate flows and exergetic efficiencies of the ORC cycle.

**Figure 5 entropy-24-01832-f005:**
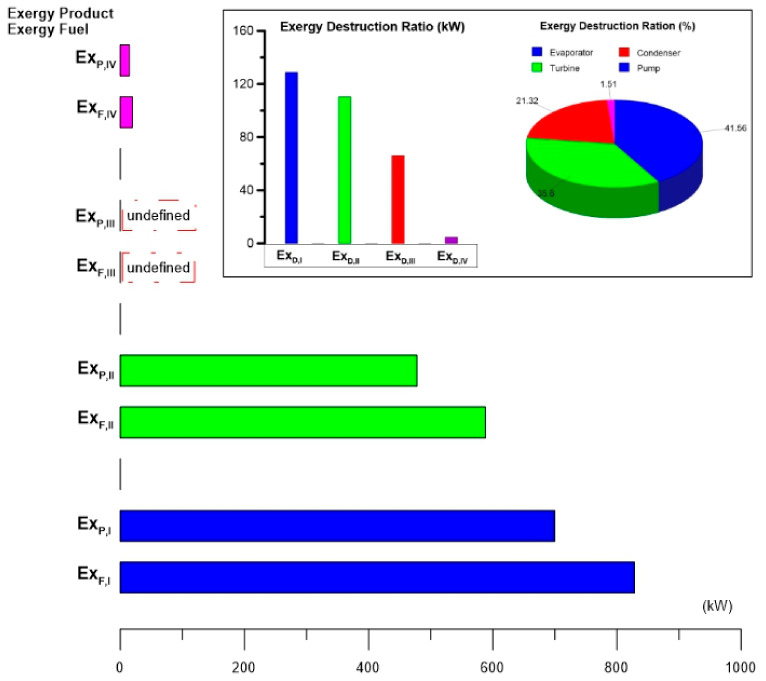
Exergy destruction rate of the components of the ORC cycle.

**Figure 6 entropy-24-01832-f006:**
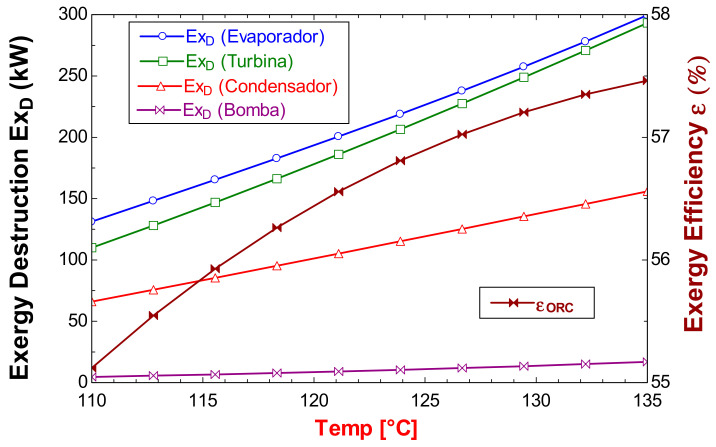
Behavior of exergy destruction rate and exergetic efficiency of the ORC.

**Figure 7 entropy-24-01832-f007:**
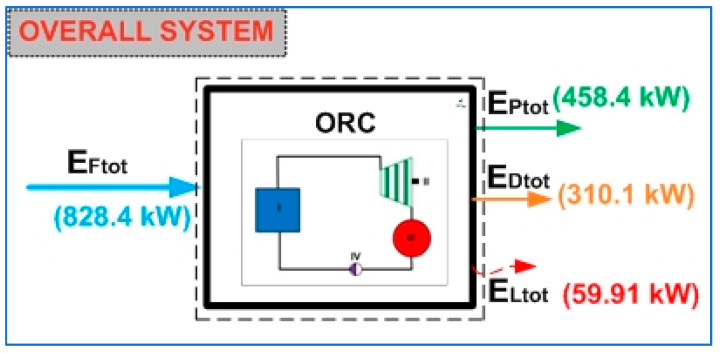
Overall exergy rate flows of the ORC cycle.

**Figure 8 entropy-24-01832-f008:**
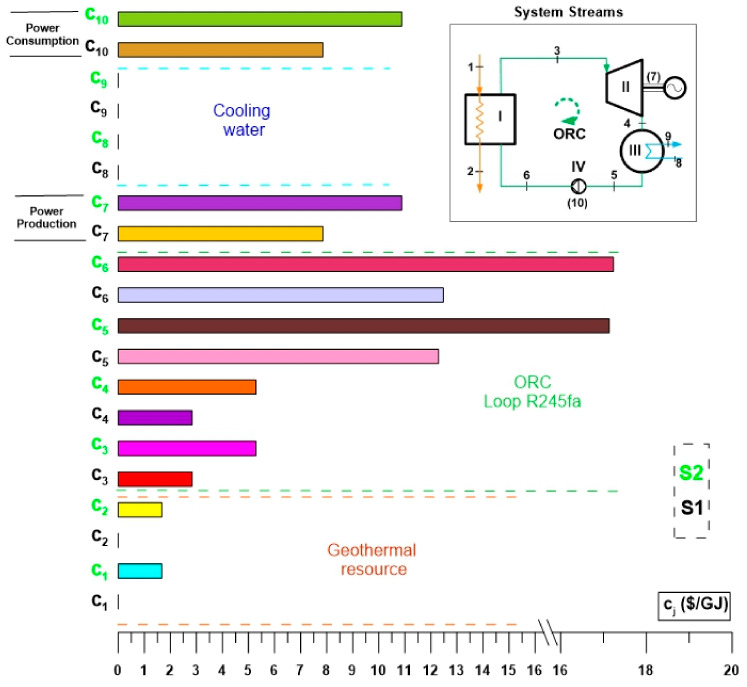
Specific costs per exergy unit of the ORC cycle in $/GJ.

**Figure 9 entropy-24-01832-f009:**
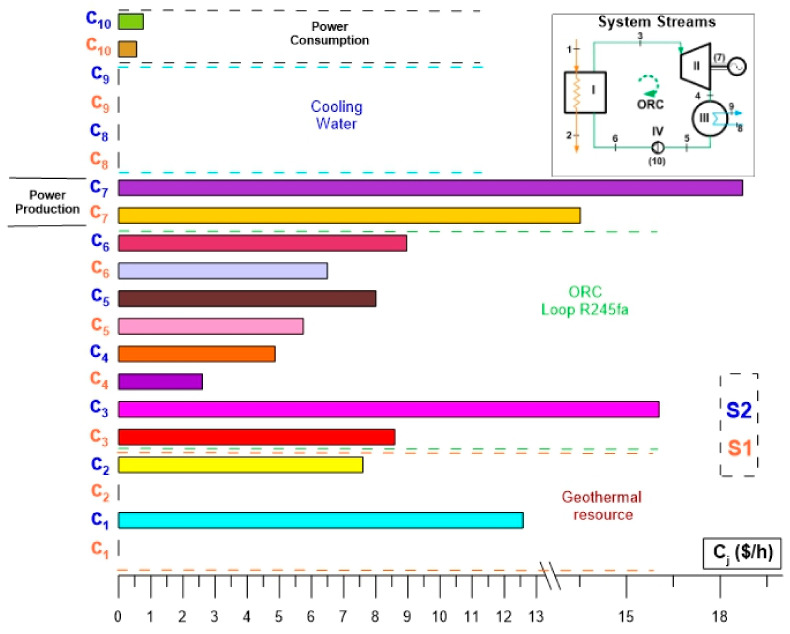
Cost rates in the ORC cycle in $/h.

**Figure 10 entropy-24-01832-f010:**
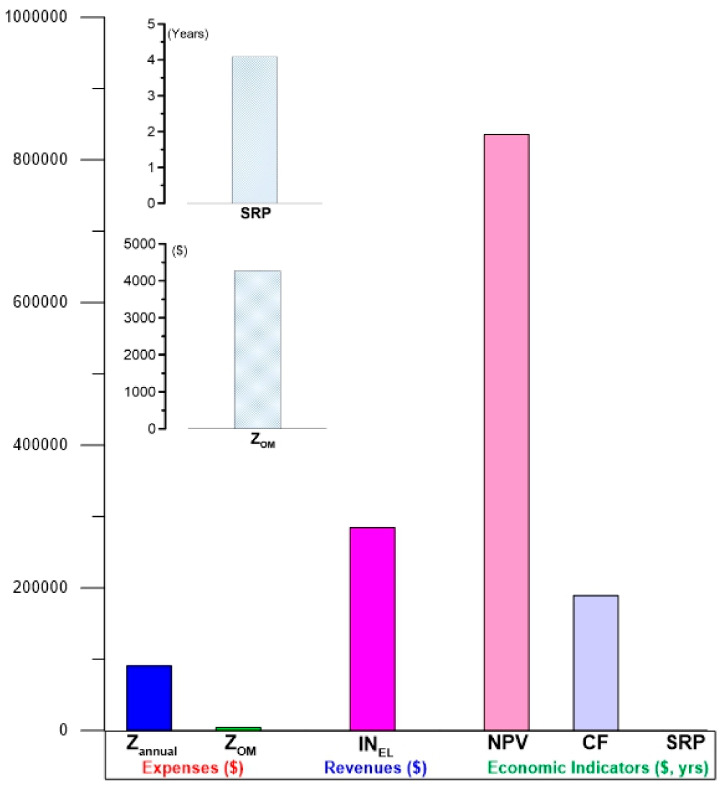
Economic feasibility results (Alternative S1).

**Figure 11 entropy-24-01832-f011:**
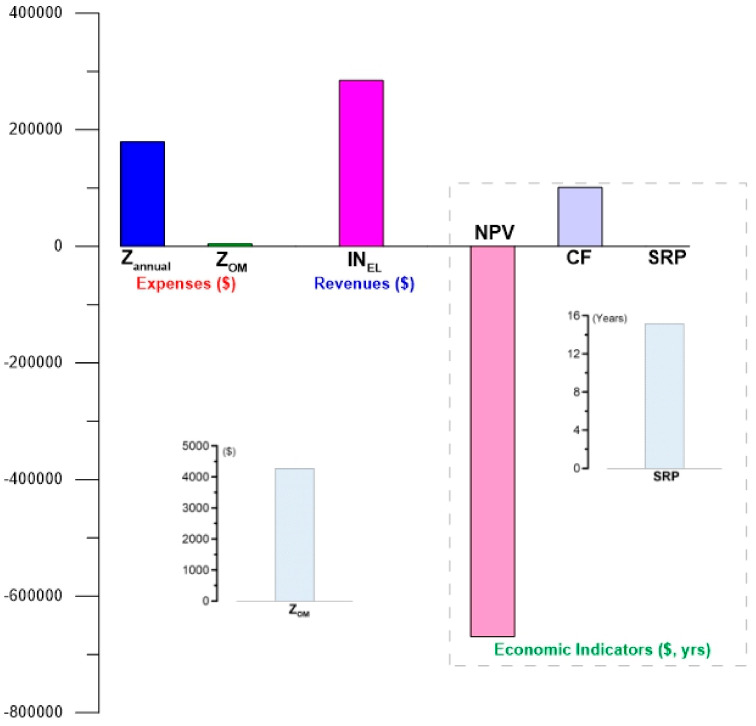
Economic feasibility results (Alternative S2).

**Figure 12 entropy-24-01832-f012:**
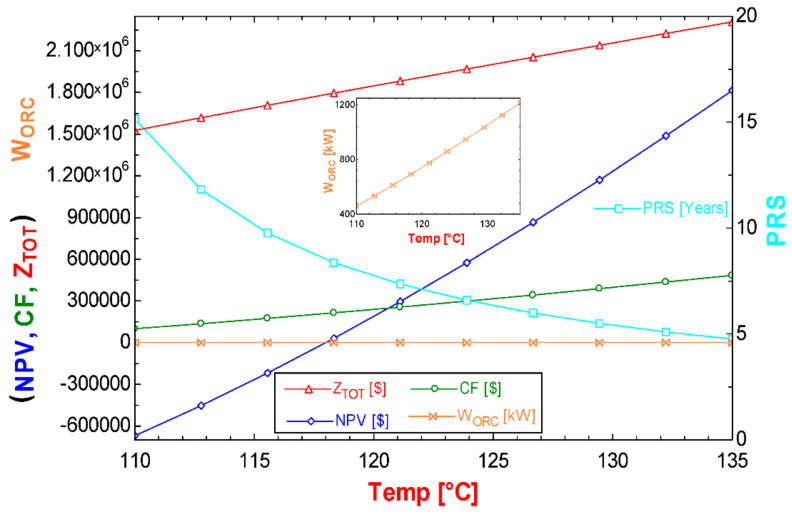
Analysis of the economic feasibility of alternative S2.

**Figure 13 entropy-24-01832-f013:**
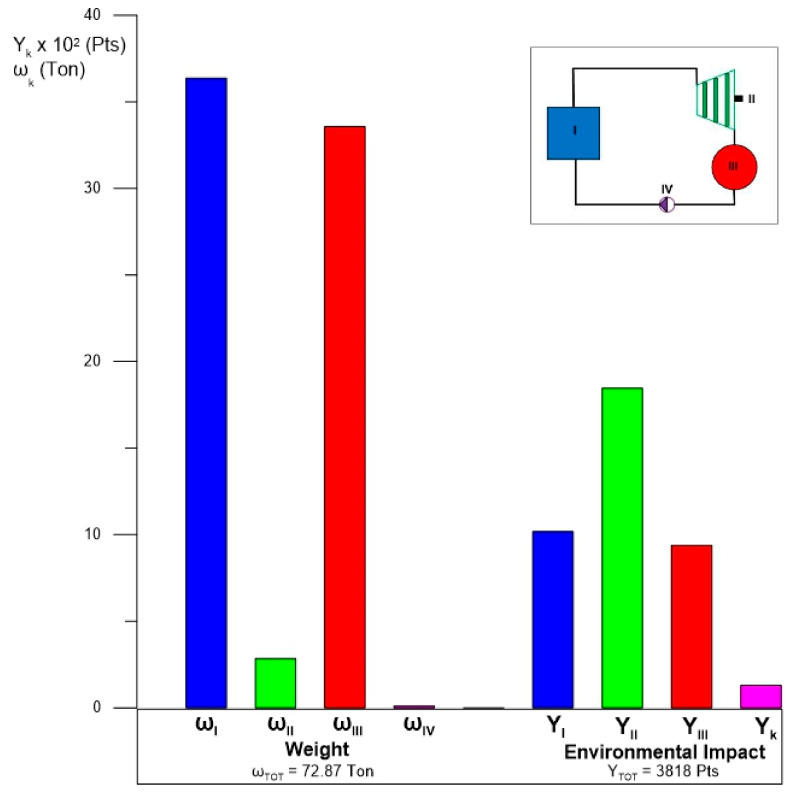
Results of the environmental impact of the components of the ORC cycle.

**Figure 14 entropy-24-01832-f014:**
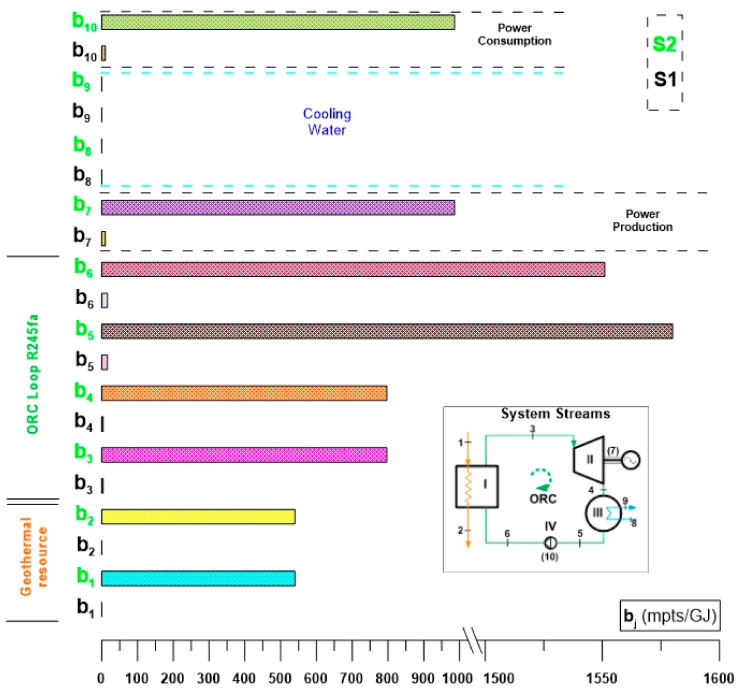
Environmental impacts per exergy unit of the ORC cycle in mpts/GJ.

**Figure 15 entropy-24-01832-f015:**
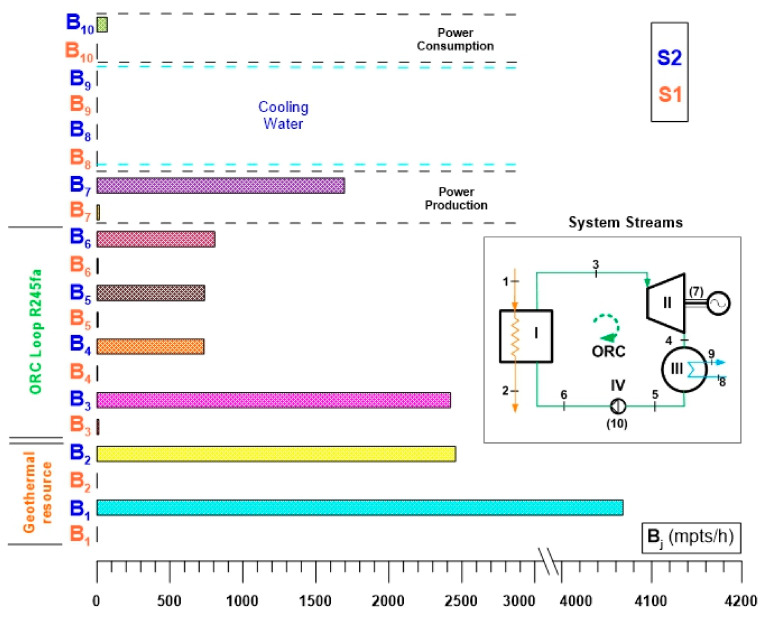
Environmental impact rate of the ORC cycle streams in (mpts/h).

**Table 1 entropy-24-01832-t001:** Background of the research work.

Refs.	Assess.	Summary
[8,9,10,11]	2E	The energetic and exergetic performance of geothermal power plants in different configurations and the integration of the ORC cycle activated with waste heat in traditional configurations have been investigated. Such a combination is well known from the concept of binary-flash geothermal power plants.
[14,15,16,17]	3E (Type I)	The energetic, exergetic and exergoeconomic feasibility of geothermal configurations, such as flash–binary geothermal power plants, has been analyzed using the ORC cycle as a binary subsystem in some cases. From the exergoeconomic point of view, fundamental parameters such as component investment costs and equipment operation and maintenance costs have been included to determine the feasibility of thermal systems.
[19]	3E (Type II)	Based on energy and exergy analysis and the combination of environmental concepts, an exergoenvironmental model of an ORC to obtain the environmental impacts of the cycle components was established.
[20,21,22,23,24]	4E	They have carried out analyses to evaluate thermal systems such as flash–binary geothermal power plants and the ORC cycle from a energetic, exergetic, exergoeconomic and exergoenvironmental way. In the exergoenvironmental evaluation they have considered indicators such as the difference in relative environmental impact and the exergoenvironmental factors. In some cases, the ORC has been remarked as the thermodynamic cycle that achieves the best performance when integrated with the flash power geothermal plant.

Assessment: 2E—Energy and Exergy; 3E (Type I)—Energy, Exergy and Exergoeconomy; 3E (Type II)—Energy, Exergy and Exergoenvironment; 4E—Energy, Exergy, Exergoeconomy and Exergoenvironment.

**Table 2 entropy-24-01832-t002:** Flash-binary geothermal power plant (T-s diagrams).

Single Flash	Binary Cycle
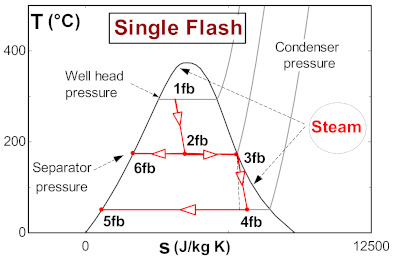	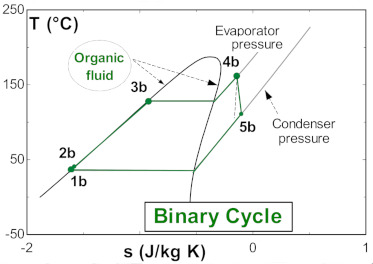

**Table 3 entropy-24-01832-t003:** ORC component diagram and temperature–entropy diagram.

Organic Rankine Cycle	T-s Diagram
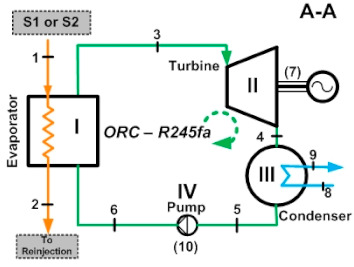	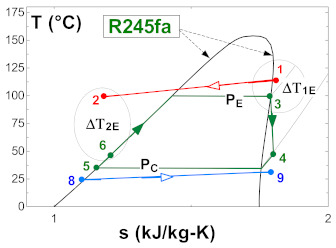

**Table 4 entropy-24-01832-t004:** ORC activation alternatives.

Alternatives	T-S Diagram
S1	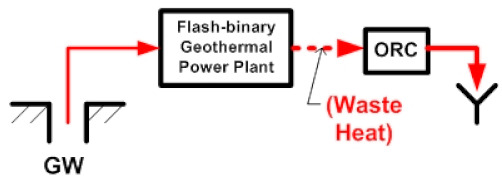
S2	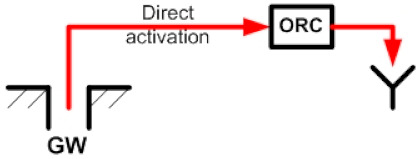

**Table 5 entropy-24-01832-t005:** Mass balances of the ORC cycle.

Variable	Equation by Component			
	Evaporator	Turbine	Condenser	Pump
Mass flow	${\dot{m}}_{6}={\dot{m}}_{3}={\dot{m}}_{r245fa}$ ${\dot{m}}_{1}={\dot{m}}_{2}$	${\dot{m}}_{3}={\dot{m}}_{4}$	${\dot{m}}_{4}={\dot{m}}_{5}$ ${\dot{m}}_{8}={\dot{m}}_{9}$	${\dot{m}}_{5}={\dot{m}}_{6}$

**Table 6 entropy-24-01832-t006:** Energy balances and efficiency parameters.

Component/Cycle	Energy	Efficiency
Evaporator	${\dot{Q}}_{I}={\dot{m}}_{r245fa}\cdot \left(h_{3}-h_{6}\right)$ ${\dot{Q}}_{I}={\dot{m}}_{geo}\cdot \left(h_{1}-h_{2}\right)$	/
Turbine	${\dot{W}}_{II}={\dot{m}}_{r245fa}\cdot \left(h_{3}-h_{4}\right)$	$\eta =\frac{h_{3}-h_{4}}{h_{3}-h_{4s}}$
Condenser	${\dot{Q}}_{III}={\dot{m}}_{r245fa}\cdot \left(h_{4}-h_{5}\right)$ ${\dot{Q}}_{III}={\dot{m}}_{h_{2}o}\cdot \left(h_{4}-h_{5}\right)$	/
Pump	${\dot{W}}_{IV}={\dot{m}}_{r245fa}\cdot \left(h_{6}-h_{5}\right)$	$\eta =\frac{h_{6s}-h_{5}}{h_{6}-h_{5}}$
ORC Cycle	${\dot{W}}_{ORC}={\dot{W}}_{II}-{\dot{W}}_{IV}$	$\eta =\frac{{\dot{W}}_{ORC}}{{\dot{Q}}_{I}}$

**Table 7 entropy-24-01832-t007:** Heat transfer parameters.

Comp.	*T-Q* Diagram	LMTD	Area
I	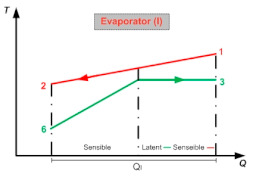	$LMTD_{I}=\frac{\left(T_{1}-T_{3}\right)-\left(T_{2}-T_{6}\right)}{ln\left(\frac{T_{1}-T_{3}}{T_{2}-T_{6}}\right)}$	$A_{I}=\frac{{\dot{Q}}_{I}}{U_{I}\cdot LMTD_{I}}$
III	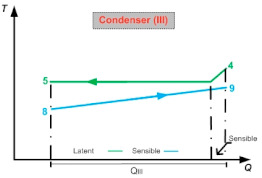	$LMTD_{III}=\frac{\left(T_{1}-T_{3}\right)-\left(T_{2}-T_{6}\right)}{ln\left(\frac{T_{1}-T_{3}}{T_{2}-T_{6}}\right)}$	$A_{III}=\frac{{\dot{Q}}_{III}}{U_{III}\cdot LMTD_{III}}$

**Table 8 entropy-24-01832-t008:** Exergy destruction rate balances and exergetic efficiencies.

Comp. Cycle	Fuel	Product	Exergy Balance	Exergetic Efficiency
I	$\dot{E}x_{1}-\dot{E}x_{2}$	$\dot{E}x_{3}-\dot{E}x_{6}$	${\dot{Ex}}_{D,I}={\left(\dot{E}x_{1}-\dot{E}x_{2}\right)}_{F}-{\left(\dot{E}x_{3}-\dot{E}x_{6}\right)}_{P}$	$\varepsilon_{I}=\frac{\dot{E}x_{3}-{\dot{Ex}}_{6}}{{\dot{Ex}}_{1}-{\dot{Ex}}_{2}}$
II	$\dot{E}x_{3}-\dot{E}x_{4}$	$\dot{E}x_{7}$	${\dot{Ex}}_{D,II}={\left(\dot{E}x_{3}-\dot{E}x_{4}\right)}_{F}-{\left(\dot{E}x_{7}\right)}_{P}$	$\varepsilon_{II}=\frac{{\dot{Ex}}_{7}}{{\dot{Ex}}_{3}-{\dot{Ex}}_{4}}$
III	/	/	${\dot{Ex}}_{D,III}={\dot{Ex}}_{4}+{\dot{Ex}}_{8}-\dot{E}x_{5}-{\dot{Ex}}_{9}$	/
IV	$\dot{E}x_{10}$	$\dot{E}x_{6}-\dot{E}x_{5}$	${\dot{Ex}}_{D,IV}={\left(\dot{E}x_{10}\right)}_{F}-{\left(\dot{E}x_{6}-\dot{E}x_{5}\right)}_{P}$	$\varepsilon_{IV}=\frac{{\dot{Ex}}_{6}-{\dot{Ex}}_{5}}{{\dot{Ex}}_{10}}$
ORC (Total)	$\dot{E}x_{1}-\dot{E}x_{2}$	$\dot{E}x_{7}-\dot{E}x_{10}$	${\dot{Ex}}_{D,tot}={\dot{Ex}}_{F,tot}-\dot{E}x_{P,tot}-\dot{E}x_{L,tot}$	$\varepsilon_{ORC}=\frac{\dot{E}x_{7}-{\dot{Ex}}_{10}}{{\dot{Ex}}_{1}-{\dot{Ex}}_{2}}$

**Table 9 entropy-24-01832-t009:** Cycle investment costs.

Component/Cycle	Cost Equation
Evaporator	$Z_{I}=2000\cdot A_{I}$
Turbine	$Z_{II}=6000\cdot {\dot{W}}_{II}^{0.7}$
Condenser	$Z_{III}=2000\cdot A_{III}$
Pump	$Z_{IV}=1120\cdot {\dot{W}}_{IV}^{0.8}$
Cycle ORC	$Z_{ORC}=Z_{I}+Z_{II}+Z_{III}+Z_{IV}$

*Z* in USD, *A* in m^2^, *W* in kW.

**Table 10 entropy-24-01832-t010:** Exergoeconomic balance equations and auxiliary equations.

Component	Cost Balance Equation		Auxiliary Equation
		S1	S2
Geothermal well (S2)	${\dot{C}}_{F}+{\dot{Z}}_{GW}={\dot{C}}_{1}$	/	${\dot{C}}_{F}=0$
Evaporator	${\dot{C}}_{1}+{\dot{C}}_{6}+{\dot{Z}}_{I}={\dot{C}}_{2}+{\dot{C}}_{3}$	${\dot{C}}_{1}=0$	$c_{1}=c_{2}$
		$c_{1}=c_{2}$	
Turbine	${\dot{C}}_{3}+{\dot{Z}}_{II}={\dot{C}}_{4}+{\dot{C}}_{7}$	$c_{3}=c_{4}$	$c_{3}=c_{4}$
Condenser	${\dot{C}}_{4}+{\dot{C}}_{8}+{\dot{Z}}_{III}={\dot{C}}_{5}+{\dot{C}}_{9}$	$c_{8}=0$	$c_{8}=0$
		$c_{8}=c_{9}$	$c_{8}=c_{9}$
Pump	${\dot{C}}_{5}+{\dot{C}}_{10}+{\dot{Z}}_{IV}={\dot{C}}_{6}$	$c_{7}=c_{10}$	$c_{7}=c_{10}$

**Table 11 entropy-24-01832-t011:** Equations for economic feasibility.

Economic Parameter	Parameter Type	Equation
Annualized investment cost	/	$Z_{anual}=Z_{TOT}\cdot \left[\frac{i\cdot {\left(1+i\right)}^{n}}{{\left(1+i\right)}^{n}-1}\right]$
Operation and maintenance cost	/	$Z_{O\&M}={\dot{W}}_{ORC}\cdot t_{op}\cdot C_{O\&M}$
Income from electricity sales	/	$IN_{EL}={\dot{W}}_{ORC}\cdot t_{op}\cdot C_{U/EL}$
Annual benefit	Feasibility indicator	$CF=\left(IN_{EL}\right)-\left(Z_{annual}-Z_{O\&M}\right)$
Net present value	Feasibility indicator	$NPV=CF\left[\frac{{\left(1+i\right)}^{n}-1}{i\cdot {\left(1+i\right)}^{n}}\right]-Z_{TOT}$
Simple return payback	Feasibility indicator	$SRP=\frac{Z_{TOT}}{CF}$

**Table 12 entropy-24-01832-t012:** Weight equations for components and environmental impact.

Component	Equation for Weight	Environmental Impact/Weight Unit
Evaporator	$ton,MW,\omega_{III}=13.91\cdot {\left({\dot{Q}}_{III}\right)}^{0.68}$	$\frac{mpts}{kg},\;bm=28$
Turbine	$ton,MW,\omega_{II}=4.90\cdot {\left({\dot{W}}_{II}\right)}^{0.73}$	$\frac{mpts}{kg},\;bm=646$
Condenser	$ton,MW,\omega_{III}=13.91\cdot {\left({\dot{Q}}_{III}\right)}^{0.68}$	$\frac{mpts}{kg},\;bm=28$
Pump	$ton,kW,\omega_{IV}=0.0631\cdot ln\left({\dot{W}}_{IV}\right)-0.197$	$\frac{mpts}{kg},\;bm=132.8$

**Table 13 entropy-24-01832-t013:** Exergoenvironmental balance equations and auxiliary equations.

Component	Exergoenvironmental Balance	Auxiliary Equation	
		S1	S2
Geothermal well (S2)	${\dot{B}}_{F}-{\dot{B}}_{1}+{\dot{Y}}_{GW}=0$	/	${\dot{B}}_{F}=0$
Evaporator	${\dot{B}}_{1}+{\dot{B}}_{6}-{\dot{B}}_{2}-{\dot{B}}_{3}+{\dot{Y}}_{I}=0$	${\dot{B}}_{1}=0$	$b_{1}=b_{2}$
		$b_{1}=b_{2}$	
Turbine	${\dot{B}}_{3}-{\dot{B}}_{4}-{\dot{B}}_{7}+{\dot{Y}}_{II}=0$	$b_{3}=b_{4}$	$b_{3}=b_{4}$
Condenser	${\dot{B}}_{4}+{\dot{B}}_{8}-{\dot{B}}_{5}-{\dot{B}}_{9}+{\dot{Y}}_{III}=0$	$b_{8}=0$	$b_{8}=0$
		$b_{8}=b_{9}$	$b_{8}=b_{9}$
Pump	${\dot{B}}_{5}+{\dot{B}}_{10}-{\dot{B}}_{6}+{\dot{Y}}_{IV}=0$	$b_{10}=b_{7}$	$b_{10}=b_{7}$

**Table 14 entropy-24-01832-t014:** Environmental evaluation equations.

Concept	Equation
Environmental impact per exergy unit of Fuel.	$bF_{k}=\frac{{\dot{B}}_{F,k}}{\dot{E}x_{F,k}}$
Environmental impact per exergy unit of the Product.	$bP_{k}=\frac{{\dot{B}}_{P,k}}{\dot{E}x_{P,k}}$
Environmental impact of exergy destruction rate.	${\dot{B}}_{D,k}=bF_{k}\cdot \dot{E}x_{D,k}$
Environmental impact of exergy losses rate.	${\dot{B}}_{L,tot}=bF_{tot}\cdot \dot{E}x_{L,tot}$
Relative difference in environmental impact.	$r_{b,k}=\frac{bP_{k}-bF_{k}}{bF_{k}}$
Exergoenvironmental factor.	$f_{b,k}=\frac{{\dot{Z}}_{k}}{{\dot{Z}}_{k}+\left({\dot{B}}_{D,k}+{\dot{B}}_{L,k}\right)}$

**Table 15 entropy-24-01832-t015:** Thermophysical properties of thermodynamic states.

#	T (°C)	P (kPa)	h (kJ/kg)	s (kJ/kg K)	x (-)	ṁ (kg/s)
1	110	-	461.3	1.419	-	48.7
2	90	-	376.9	1.193	-	48.7
3	100	1269	474.1	1.791	1	18.09
4	52.87	211	447.7	1.811	-	18.09
5	35	211	245.8	1.157	0	18.09
6	35.6	1269	246.9	1.158	-	18.09
8	25	-	104.8	0.3669	-	87.31
9	35	-	146.7	0.5049	-	87.31

**Table 16 entropy-24-01832-t016:** Energy analysis results.

Energy Flows (kW)	Energy Distribution (%)
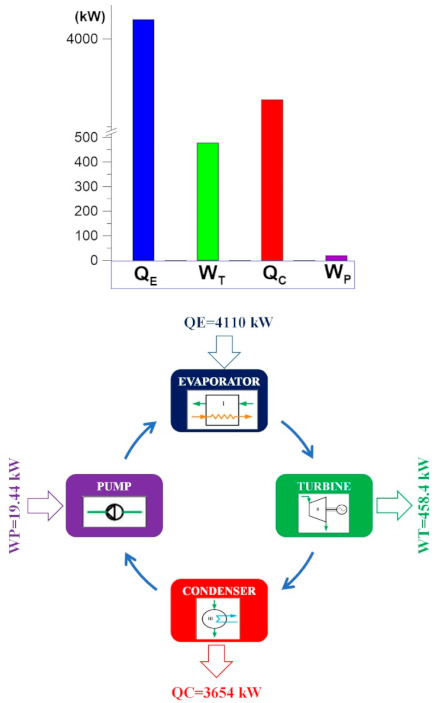	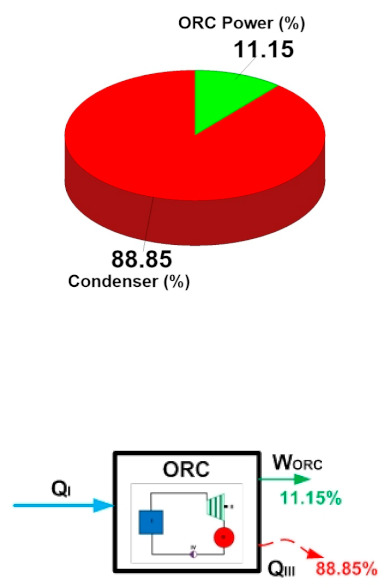

**Table 17 entropy-24-01832-t017:** Results of heat transfer parameters.

Component	LMTD (°C)	Area (m^2^)
Evaporator	26.21	62.72
Condenser	13.56	93.86

**Table 18 entropy-24-01832-t018:** ORC investment costs.

Investment Cost ($)	Investment Cost Ratio (%)	
	S1	S2
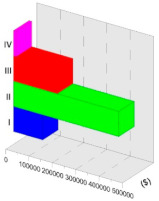	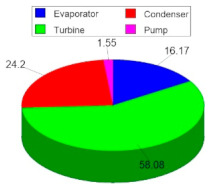	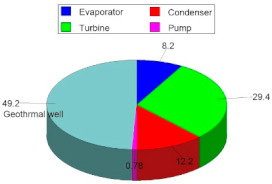

**Table 19 entropy-24-01832-t019:** Results of the thermoeconomic evaluation.

Variable	Units	Alternative	Component			
			Evaporator	Turbine	Pump	Condenser
$cF_{k}$	($/GJ)	S1	0	2.82	7.85	/
		S2	1.67	5.28	10.88	/
$cP_{k}$	($/GJ)	S1	0.83	7.85	14.15	/
		S2	2.81	10.88	18.13	/
${\dot{C}}_{D,k}$	($/h)	S1	0	1.123	0.132	/
		S2	0.775	2.098	0.183	/
${\dot{C}}_{L,k}$	($/s)	S1	/	/	/	0
		S2	/	/	/	0.36
$r_{k}$	(%)	S1	infinity	178	80.05	/
		S2	68.26	106	66.67	/
$f_{k}$	(%)	S1	100	87.02	60.22	/
		S2	73.01	78.21	52.24	/

**Table 20 entropy-24-01832-t020:** Results of the exergoenvironmental evaluation.

Variable	Units	Alternative	Component			
			Evaporator	Turbine	Pump	Condenser
$bF_{k}$	(mpts/GJ)	S1	0	4.146	9.521	/
		S2	540	797	986	/
$bP_{k}$	(mpts/GJ)	S1	1.66	9.52	13.6	/
		S2	641	986.0	130	/
${\dot{B}}_{D,k}$	(mpts/h)	S1	0	1.65	0.161	/
		S2	250.56	316.8	16.66	/
${\dot{B}}_{L,k}$	(mpts/h)	S1	/	/	/	0
		S2	/	/	/	116.28
$r_{k}$	(%)	S1	infinity	129.6	42.9	/
		S2	18.73	23.66	31.95	/
$f_{k}$	(%)	S1	100	82.18	25.77	/
		S2	1.645	2.342	0.3342	/

## Data Availability

Data supporting the research of this work are available from the corresponding author upon reasonable request.

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
