# Peer review of "4E Assessment of an Organic Rankine Cycle (ORC) Activated with Waste Heat of a Flash–Binary Geothermal Power Plant"

_entropy, 2022, doi:10.3390/e24121832_

Round 1
Reviewer 1 Report
The article is very interesting but needs some corrections and better attention to detail.
1. The introduction should be shortened. The first paragraph is unnecessary. There are lots of descriptions below. Therefore, a more detailed description in this section is not needed. The authors should emphasize the novelty of the article without such a developed description of the cited works. For the clarity of the introduction, please include a table with the cited works and summary.
2. The T-s plot and table 13 shows the beginning of expansion in the saturation curve. Were the calculations made without overheating the working medium? If so, please explain this approach. This is far from realistic conditions where overheating of the medium should be designed for at least 5K.
3. Is the cost of the working fluid included in the work? If not, please comment. Did the authors consider a comparison for a different working medium?
4. A comment on the choice of ORC cycle type (basic) would also be useful. Why was regeneration not used?
5. Most of the figures should be improved (for example Figure 1.). Figures should be legible and transparent as in the publication, e.g.: https://www.sciencedirect.com/science/article/pii/S019689041630111X
Author Response
Thank you for your reviewing, we highly appreciate your comments and suggestions to improve the quality of our work so it can be accepted in the Journal Entropy. You can find the responses and our comments to your suggestions in the attached document.

Reviewer 2 Report
In this work, a 4E assessment is carried out for the ORC application in a geothermal power plant, and two alternatives are compared. The results illustrate the possibility for the waste heat recovery of geothermal by using ORC application. The topic is interesting and the manuscript is well organized. However, there are still some problems related to the system analysis. Therefore, the paper cannot be published in Entropy as it is presented and I recommend a major revision of this paper. Here are several comments as below:
1. In the previous work, the geothermal power plant by using ORC has developed for a long time, and the author should stress the advantages of the presented system.
2. Please explain why choosing R245fa as the working fluids.
3. The T-s diagram of the alternative S1 ORC combined flash-ORC geothermal power system is suggested to be carried out.
4. In order to improve the readability of the manuscript, please complete the ordinates of the axis in Fig. 2 and Fig. 3.
5. The paper mostly focuses on the ORC system performance, the analysis on the alternative S1 ORC benefit to the overall system performance promotion should be furtherly developed.
6. I think it is not proper to directly compare the system economic performance between the alternative S1 and S2, the S2 should be compared to the overall S1 combine flash-ORC geothermal power system.
7. The future practical application of the presented system should be further discussed.
Author Response
Thank you for your valuable opinions, comments and suggestions about this work. You can find the answers to your suggestions in the attached document: We have considered your comments to improve the quality of our work and we greatly appreciate your suggestions.

Reviewer 3 Report
The manuscript "4E assessment of an Organic Rankine Cycle (ORC) activated with waste heat of a flash-binary geothermal power plant" is an interesting article to read in which authors analyze the research gap of the thermodynamic performance of the ORC cycle activated with low-temperature waste heat from flash-binary geothermal power plants from the perspectives of the energetic, exergetic, exergoeconomic and exergoenvironmental performance analysis of its configuration.
The innovation of this work focuses on the recovery of that energy from that waste heat, proposing the integration of a low-temperature ORC in a flash-binary geothermal power plant and analyzing that recovery to demonstrate the 136 thermodynamic, economic and environmental performance. In the same way, the comparison of the recovery of waste heat against the direct activation of the ORC cycle through low-temperature geothermal wells is proposed, to analyze the advantages and disadvantages of both configurations, as well as their viability in practical application.
The manuscript has been displayed in a good manner and also written in a very technical way. However I have minor comments related to the manuscript:
1. The English needs to be rechecked as there are some grammatical mistakes in the manuscript.
2. I have not seen the experimental/theoretical validation of the developed model. So it is highly recommended to have the validation of the model before proceeding for the acceptance of the manuscript.
Author Response
Thank you for your valuable opinions and suggestions on this work. We have considered your comments to improve the quality of our work and achieve acceptance in the Journal entropy. We greatly appreciate your suggestions.

Round 2
Reviewer 1 Report
Thank you for responding to my comments and I suggest the accpetance of this paper. In the next works, please assume parameters as close to real parameters as possible and take into account all necessary costs.
Reviewer 2 Report
The revised paper has addressed all the comments, thus the paper can be published in this journal.
Reviewer 3 Report
I agree with the revision made.